# Long-range order enhance performance of patterned blue quantum dot light-emitting diodes

Yuyu Jia[1,2], Hui Li [3] ✉, Ning Guo[2,4], Fengmian Li[2,5], Tianchen Li[2,4], Haoran Ma[1], Yuyan Zhao[3], Hanfei Gao[3], Dan Wang[1], Jiangang Feng [3], Zhiyuan He [1] ✉, Lei Jiang [2,3] & Yuchen Wu [2,3,4] ✉

Quantum dot light-emitting diodes show great potential for next-generation displays. Although film quantum dot light-emitting diodes have achieved or approached efficiency and stability standards for commercial applications, patterned quantum dot light-emitting diodes, particularly blue quantum dot light-emitting diodes, still face challenges in both efficiency and stability. Traditional patterning methods often lead to defects and disorder, causing non-radiative recombination and reduced performance. Here, we develop an aromatic-enhanced capillary bridge confinement strategy to achieve long-range ordered blue quantum dot microstructure arrays. These quantum dot arrays integrate into quantum dot light-emitting diodes achieve a peak external quantum efficiency of 24.1% and a peak luminance of 101,519 cd m$^{-2}$. Additionally, the minimum pixel size is reduced to 3 μm, enabling a maximum resolution exceeding 5000 pixels per inch, and static electroluminescence display modes. This study provides a strategy to advance the commercialization of quantum dot light-emitting diodes.

Quantum dot light-emitting diodes (QLEDs) demonstrate characteristics such as reduced energy requirements, narrow-band emission spectra, broad coverage of the visible color spectrum, and high contrast[1–9], making them a strong contender for human-machine interface display technologies. Currently, the highest external quantum efficiencies (EQE) for red, green, and blue film QLEDs reach 38.2%, 29.2% and 23.5%, respectively and operational lifetimes ($T_{95}$, measured as the duration until luminance decays to 95% of initial value $L_0$) exceeding 48,000 h, 15,600 h, and 227 h, respectively[8,10–13]. However, patterned QLEDs, especially blue patterned QLEDs, generally exhibit lower efficiency, luminance, and poorer stability, posing challenges to the commercialization of full-color QLEDs. Hence, developing blue QLEDs with higher efficiency, luminance, stability, and resolution is vital for developing viable pathways for industrial fabrication of full-color quantum dot (QD) display panels.

Compared to red and green QDs, the inherently smaller particle size of blue QDs results in weaker interactions between them. This makes it challenging to overcome the complex fluid dynamics during inkjet printing, which affects the long-range ordered self-assembly of QDs. Integrating disordered QD microstructure arrays into QLEDs increases the risk of charge leakage and non-radiative recombination during operation[14,15]. Moreover, the larger specific surface area of blue QDs allows long-chain fatty acid ligands like oleic acid (OA) and oleylamine to occupy more space in the film, thereby weakening the effective transport of charge carriers[16,17]. Therefore, developing effective strategies to enhance the interaction between blue QDs and

[1]School of Materials Science and Engineering, Beijing Institute of Technology, Beijing, PR China. [2]Key Laboratory of Bio-inspired Materials and Interfacial Science, Technical Institute of Physics and Chemistry, Chinese Academy of Sciences, Beijing, PR China. [3]State Key Laboratory of Bioinspired Interfacial Materials Science, Suzhou Institute for Advanced Research, University of Science and Technology of China, Suzhou, Jiangsu, PR China. [4]University of Chinese Academy of Sciences (UCAS), Beijing, PR China. [5]Key Laboratory of Bio-Inspired Smart Interfacial Science and Technology of Ministry of Education, School of Chemistry, Beihang University, Beijing, PR China. ✉e-mail: lihui17703806212@163.com; hezy@bit.edu.cn; wuyuchen@iccas.ac.cn

achieve their long-range ordered self-assembly is crucial for improving the efficiency of charge carrier transport. The aromatic ligands with shorter chain lengths have been proven to enhance carrier mobility and conductivity[18,19]. Additionally, the π-π interactions between aromatic ligands can strengthen the attraction between QDs, aiding their long-range ordered assembly and thereby improving the overall performance of the QLEDs[20].

Here, we develop an aromatic-enhanced capillary bridge confinement strategy for assembling a long-range ordered array of blue QD microstructures. Initially, aromatic ligands (3-fluorocinnamate, 3-F-CA) are introduced onto the QD surfaces to bolster interparticle interactions. Subsequently, employing micropillar templates, the blue QD liquid film with aromatic ligands is uniformly segmented. Through controlled directional motion of the three-phase contact lines (TPCLs) within isolated capillary bridges, we successfully fabricated highly ordered arrays of blue QD microstructures. When incorporated into QLED devices, these arrays demonstrate favorable electrical performance: (i) achieving a minimum pixel size reduced to 3 μm, yielding a resolution exceeding 5000 Pixels Per Inch (PPI); (ii) achieving static electroluminescent (EL) pattern display with a pixel size of 5 μm; (iii) attaining peak external quantum efficiency (EQE) of 24.1% and maximum luminance surpassing $1 \times 10^5$ cd m$^{-2}$; (iv) nearing commercial application lifespans (extrapolated $T_{95}$ lifetime of 54 h at 1000 cd m$^{-2}$). This advancement in blue QD self-assembly technology enhances the efficiency, luminance, stability, and scalability of blue patterned QLEDs, advancing the feasibility of industrial production for full-color QLED displays.

## Results

### Preparation and characterization of quantum dot arrays

Van der Waals forces are important interactions affecting the self-assembly of nanocrystals[14], with their magnitude dependent on QD size. Owing to their smaller physical dimensions, blue QDs exhibit weaker interparticle attraction than red and green QDs. The inkjet-printed QD arrays lead to the formation of numerous defects and disordered regions[21]. These defects and disordered areas increase QD quenching and elevate charge leakage (Fig. 1a). Such issues significantly reduce both the optoelectronic conversion efficiency and the stability of the patterned device. Therefore, the quality of blue QD arrays is crucial for the overall performance of patterned QLEDs. We achieve long-range ordered assembly of QDs through capillary bridge confinement, effectively addressing the issues encountered in QLED operation (Fig. 1b), thereby significantly enhancing QLED performance[22]. To better achieve long-range ordered configurations of blue QD microstructures, it is crucial to enhance the interaction forces between blue QDs. Compared to long-chain fatty acid ligands (such as oleic acid), short-chain aromatic ligands can not only improve carrier mobility[23] but also enhance the attraction between QDs through π-π interactions, which helps overcome the interference of complex hydrodynamics on the self-assembly process of blue QDs[24].

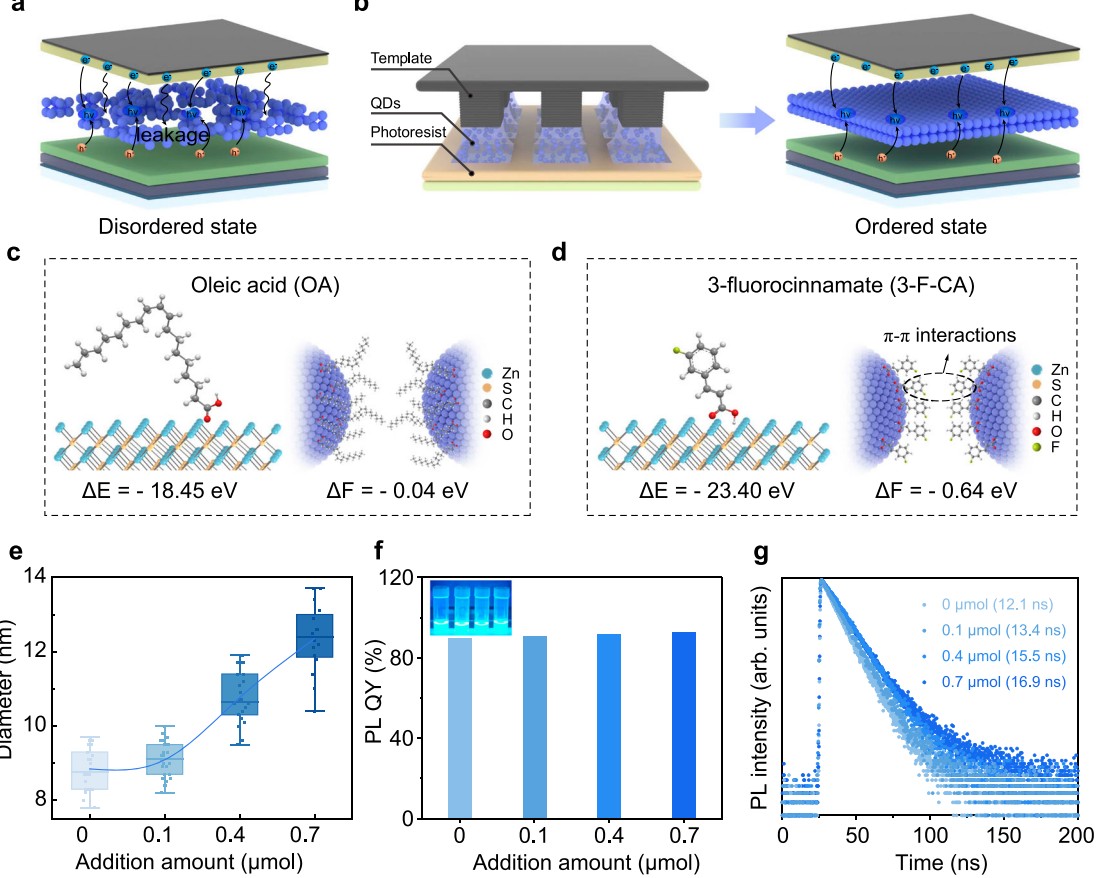

**Fig. 1 | The effect of 3-F-CA on the performance of QDs. a** The disordered QD layer leads to charge leakage. **b** Schematic diagram of long-range ordered arrangement of QDs under capillary bridge confinement (left). The ordered QD layer is uniformly dense, suppressing charge leakage (right). **c** The schematic diagram of the binding energy (ΔE, left) during surface modification of QDs with OA and the interaction energy (ΔF, right) between adjacent QDs. **d** The schematic diagram of the binding energy (left) during surface modification of QDs with 3-F-CA and the interaction energy (right) between adjacent QDs. **e** DLS volume-based size distribution graphs of QDs ligand exchange with the different concentration of the 3-F-CA (0, 0.1, 0.4, 0.7 μmol). Error bars represent the standard deviation of twenty independent samples. **f** The PLQY of the QD solution after addition of varying amounts of 3-F-CA (0, 0.1, 0.4, 0.7 μmol). Inset, photograph taken under illumination at 365 nm; from left to right: 0, 0.1, 0.4, 0.7 μmol. **g** The transient PL decay of the QD solution after addition of varying amounts of 3-F-CA (0, 0.1, 0.4, 0.7 μmol).

To prepare QD microstructures with long-range ordered configurations, we synthesized oleic acid-modified blue QDs with the crystal structure CdZnSe/ZnSe/ZnSeS/ZnS[7-9,25-27] (see details of synthesis in "Methods"). The synthesized blue QDs exhibit a photoluminescence (PL) emission peak at 472 nm, a full-width at half-maximum (FWHM) of 28 nm, and a high photoluminescence quantum yield (PLQY) of up to 90% (Supplementary Fig. 1a). Transient PL decay results indicate that the lifetime of the blue QDs in solution is nearly single-exponential, with a value of 12.1 ns (Supplementary Fig. 1b). Transmission electron microscope (TEM) images (Supplementary Fig. 2a, b) reveal that the synthesized blue QDs are well-dispersed with uniform particle sizes. The core has an average diameter of 4 nm (Supplementary Fig. 2d), and the size increases to 8 nm after coating with the shell (Supplementary Fig. 2e). Perfect lattice fringes are observed in the HAADF-STEM image of the blue QDs (Supplementary Fig. 2c), indicating the formation of excellent crystallinity during the epitaxial shell growth process. X-ray diffraction (XRD) patterns (Supplementary Fig. 2f) also confirm that the blue QDs possess a zincblende structure, with characteristic Bragg peaks at (100), (220) and (311).

To enhance the interaction between blue QDs, we introduce 3-F-CA to the surface of the QDs (see details in "Methods"). The element distribution (Supplementary Fig. 3) clearly demonstrates the continuous gradient core-shell structure of the QDs. S elements and F elements are mainly distributed in the outermost layer. Hydrogen signals from 3-F-CA are exclusively observed in the $^1$H nuclear magnetic resonance (NMR) spectra of 3-F-CA-modified QDs (Supplementary Fig. 4). These indicate that 3-F-CA has successfully bound to the QDs. Density functional theory (DFT) calculations are employed to explore the impact of ligands on the interactions between QDs. Compared to the long-chain OA (Fig. 1c, left), 3-F-CA ligand exhibits a higher binding energy (Fig. 1d, left and Supplementary Fig. 5), suggesting that 3-F-CA can more effectively passivate the QD. The interaction energy ($\Delta F$) between OA modified QDs is −0.04 eV (Fig. 1c, right and Supplementary Fig. 6a), while $\Delta F$ between 3-F-CA modified QDs is −0.64 eV (Fig. 1d, right and Supplementary Fig. 6b). In contrast, the interaction between 3-F-CA modified QDs is significantly enhanced. Further details of the DFT calculations can be found in Methods.

To determine the optimal amount of 3-F-CA, we compare the optical properties of QDs with different amounts of 3-F-CA. The dynamic light scattering (DLS) diameter of the QD solution gradually increases with the addition of 3-F-CA (Fig. 1e), further demonstrating the impact of 3-F-CA on the interactions between the QDs. As the amount of 3-F-CA increases, the PLQY of the QDs gradually increases. At an addition amount of 0.7 μmol, the PLQY of the QD solution reaches 93% (Fig. 1f). The transient PL decay maintains a single-exponential profile, reaching a lifetime of 16.9 ns, which is higher than that of OA-modified QD solution (12.1 ns, Fig. 1g). This is likely due to the stronger binding ability of 3-F-CA to the QD surface, which can more effectively passivate the surface[28]. To determine the ligand density of 3-F-CA on the QDs surfaces, deconvolution analysis of the X-ray photoelectron spectroscopy (XPS) peaks is performed (Supplementary Fig. 7). Using the Zn signal associated with carboxylic acid (-COOH) groups in the QDs as a reference (Supplementary Table 1), the normalized ratio of F is determined to be 0.84. We integrated the OA modified QD layer and the 3-F-CA modified QD layer, both prepared by spin coating, into the QLED. The QLED with 3-F-CA exhibits an EQE and peak luminance of 16.4% and 71,591 cd m$^{-2}$, respectively, both of which are 2.1 times and 2.6 times higher than those of QLEDs with OA, respectively (EQE and peak luminance of 7.7% and 26,641 cd m$^{-2}$, Supplementary Fig. 8).

To elucidate the relationship between the long-range order of QDs and the interactions between them, we constructed a sandwich-like system consisting of a micropillar template[29-31], a continuous liquid film of blue QDs modified with two types of ligands, and a substrate with a microhole array (the preparation and modification methods for the micropillar template and the preparation process for the micropores are detailed in "Methods" and Supplementary Fig. 9a), as shown in Fig. 2a. Initially, the continuous liquid film is pinned to the hydrophilic top of the micropillar template, and as the solvent evaporates, the film is uniformly divided into a series of independent capillary bridges. Within the independent capillary bridges, the TPCLs undergo directed sliding. The inward Laplace pressure at the meniscus balances the pinning forces originating from the substrate and the deposited QDs[32], ultimately achieving long-range ordered arrangement of 3-F-CA-modified blue QDs within the micropores (Fig. 2a, left). This is due to the π-π interactions between the 3-F-CA ligands, which enhance the interactions between the QDs, effectively overcoming the effects of complex fluid dynamics on the long-range ordered self-assembly of the QDs. The formation of the QD microstructure arrays is observed in situ using fluorescence microscopy (Supplementary Fig. 9b). Subsequently, QD arrays were prepared using QD solutions with different concentrations (2, 5, 10, and 15 mg mL$^{-1}$). The fluorescence micrographs of the obtained QD arrays are shown in Fig. 2b, with schematic illustrations of the assembled QD microstructures displayed in the upper-left corner of each subfigure. As the solvent continues to evaporate and the TPCL shrank, QDs preferentially reach saturation and deposit in the TPCL around the micropores. At a low QD concentration (2 mg mL$^{-1}$), the QDs in the solution are insufficient, resulting in incomplete microstructure formation, manifested as insufficient QD deposition in the central region. As the QD concentration increases, the unfilled central area gradually decreases (5 mg mL$^{-1}$) until complete filling is achieved at an appropriate concentration (10 mg mL$^{-1}$). Conversely, at an excessively high concentration (15 mg mL$^{-1}$), more QDs remain in the central region, producing a brighter area in the center. Based on the above investigations, we successfully fabricate large-area blue QD arrays using 3-F-CA modified QD solution with a concentration of 10 mg mL$^{-1}$.

Fluorescence microscopy image of the large-area blue QD arrays is presented in Fig. 2c, which reveals that the fabricated QD arrays exhibit uniform blue luminescence with well-defined boundaries. In addition, the PLQY of the QD arrays reaches up to 85% under different excitation powers (Supplementary Fig. 10a). Furthermore, the transient PL decay of the 3-F-CA modified ordered QD layer remains single-exponential, with a value of 12.2 ns (Supplementary Fig. 10b). The scanning electron microscopy (SEM) images reveal that the prepared QD microstructure arrays exhibit uniform size distribution (Fig. 2d). At higher magnification, the QDs containing 3-F-CA show a clear long-range ordered arrangement, while the blue QDs without 3-F-CA appear disordered (Supplementary Fig. 11a). Atomic force microscopy (AFM) images further show that the surface of the QD microstructures containing 3-F-CA is smoother (Fig. 2e and Supplementary Fig. 12), whereas the QD microstructures without 3-F-CA exhibit higher surface roughness (Supplementary Fig. 11b). Compared to the QD microstructures without 3-F-CA (Supplementary Fig. 11c), the QD microstructures containing 3-F-CA exhibit distinct Bragg diffraction spots in the grazing incidence small-angle X-ray scattering (GISAXS) images (Fig. 2f). Analysis of the GISAXS azimuthal integration (Fig. 2g) reveals that the lattice parameter of the QD microstructures containing 3-F-CA is 9.2 nm, while the average inter-particle distance of the QD microstructures without 3-F-CA is 9.7 nm (Supplementary Fig. 11d), which is consistent with the average point-to-point distance (Supplementary Figs. 2e and 13). These characterization results provide strong evidence that the QDs containing 3-F-CA achieve long-range ordered arrangement under the influence of the micropillar template. We also replaced the OA ligands on the QD surface with 3-cyclohexylpropionic acid (3-CPA). Notably, 3-CPA modified QDs fabricated via the capillary bridge-mediated confined assembly strategy exhibit disordered packing

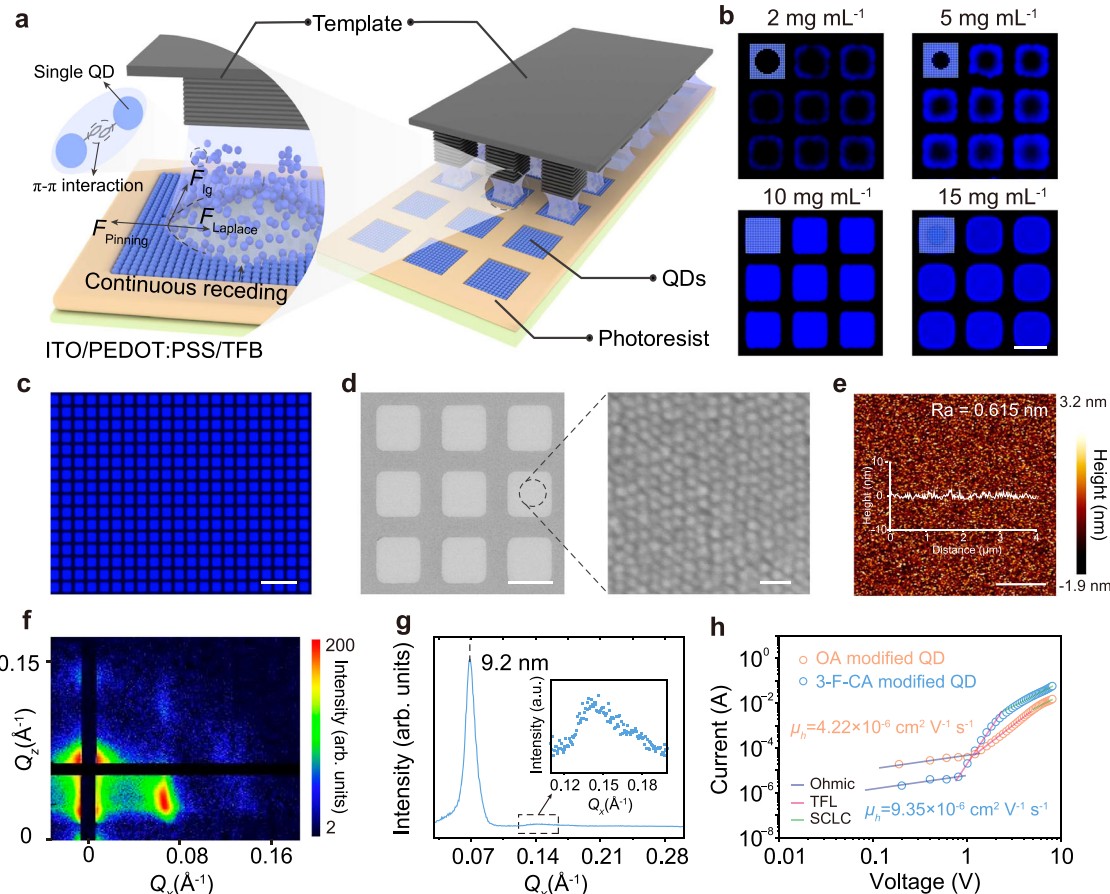

**Fig. 2 | Preparation and performance of long-range ordered QD arrays.**
**a** Laplace pressure can balance the pinning force, achieving for continuous dewetting process and regulated QDs assembly. **b** Fluorescence microscopy image of microstructures assembled from the blue QD solutions with varying concentrations (2, 5, 10, 15 mg mL⁻¹, scale bar is 5 µm). **c** Fluorescence microscopy image of the blue ordered QD microstructures arrays (scale bar is 20 µm). **d** SEM image of the ordered QD microstructures arrays (scale bar is 5 µm) and detailed image (scale bar is 20 nm). **e** AFM image of the ordered QD microstructures (scale bar is 1 µm). **f** GISAXS pattern of the ordered QD microstructures. **g** Azimuthal integration of GISAXS diffraction patterns from ordered QD microstructures. Inset, azimuthal integration of GISAXS diffraction patterns from ordered QD microstructure within the 0.11–0.20 range. **h** Space charge-limited current measurements of hole-only devices based on OA modified disordered QDs and 3-F-CA modified ordered QDs.

(Supplementary Fig. 14), further demonstrating that π-π interactions promote long-range ordered packing of QDs.

To investigate the impact of long-range ordered QD structures on charge carrier transport in devices, we fabricated hole-only devices (device structure: indium tin oxide (ITO)/PEDOT:PSS/TFB/QD arrays/MoO₃/Al, Supplementary Fig. 15a) and electron-only devices (device structure: ITO/ZnMgO/QD arrays/ZnMgO/Al, Supplementary Fig. 15b) based on QD microstructure arrays with both long-range ordered and disordered configurations. The electron and hole mobilities of both devices are calculated by fitting the space-charge-limited current (SCLC) region, assuming single trap level (see details of calculation in Supplementary Note 1)[33,34]. The results show that the hole mobility (9.35 × 10⁻⁶ cm²V⁻¹ s⁻¹, Fig. 2h) and electron mobility (1.44 × 10⁻⁴ cm²V⁻¹ s⁻¹, Supplementary Fig. 15c) of the QLED devices with long-range ordered configuration are 2.2 times and 1.6 times higher, respectively, compared to the hole mobility (4.22 × 10⁻⁶ cm²V⁻¹ s⁻¹, Fig. 2h) and electron mobility (0.90 × 10⁻⁴ cm²V⁻¹ s⁻¹, Supplementary Fig. 15c) of the QLED devices with disordered configuration. We fabricated field-effect transistors (FET) to determine the conductivity of the QD films (Supplementary Fig. 15d)[35]. The calculated conductivity of the ordered QD layer is found to be 4.2 × 10⁻⁴ S m⁻¹ (Supplementary Note 2). These results suggest that the long-range order of the QDs significantly enhances carrier transport in the devices, which is beneficial for improving the performance of QLEDs.

## Device performance characterization

Two types of QD microstructure arrays were integrated into bottom-emission QLEDs with the structure of ITO/PEDOT:PSS/TFB/QD arrays/ZnMgO/Al (Supplementary Fig. 16a−c). The calculation of the active area of the QLED is detailed in Supplementary Note 3. The QLED with long-range ordered configuration exhibits saturated blue emission, as evidenced by its color coordinates of (0.107, 0.146) (Supplementary Fig. 16d), while maintaining a stable emission wavelength across varying applied voltages (Fig. 3a). Comparative analysis of the current density-luminance-voltage characteristics for both QLED structures (Fig. 3b) reveals that the long-range ordered configuration exhibits reduced current density prior to turn-on, suggesting less charge leakage (Supplementary Fig. 17)[35]. Notably, at an equivalent current density (~4.0 V, ~12 mA cm⁻²), the ordered QLED achieves a luminance fourfold greater than its disordered counterpart. Furthermore, the peak EQE of the ordered QLED reaches 24.1% (Fig. 3c), accompanied by a maximum luminance exceeding 1 × 10⁵ cd m⁻². These values are 1.8 times and 5 times higher than those of the disordered QLED, making it one of the best-performing blue QLEDs reported to date (Supplementary Table 2 and Supplementary Fig. 18). As shown in the histogram in Fig. 3d, the average EQE of the two devices are 13.5% and 22.0%, respectively, indicating good reproducibility (Supplementary Fig. 19).

Stability tests were conducted on the long-range ordered and disordered QLEDs at nearly the same initial luminance (Fig. 3e). The

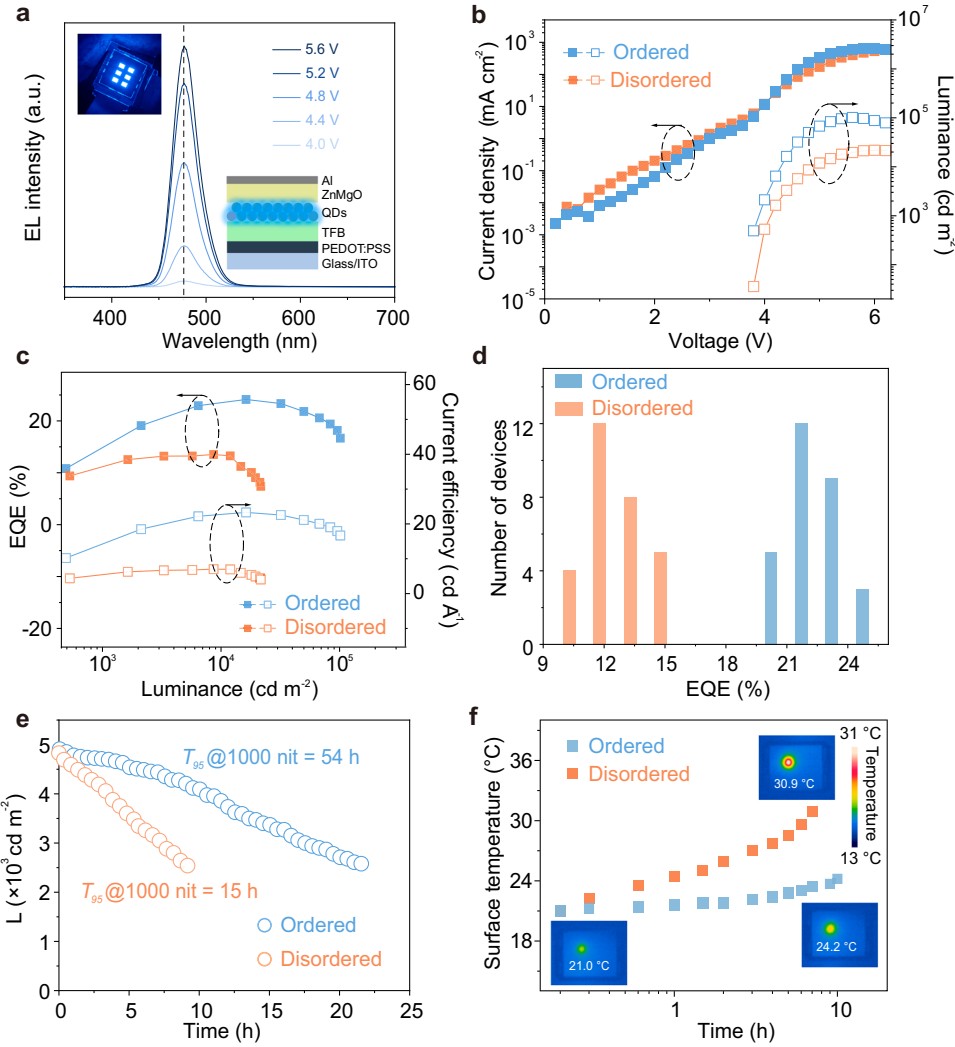

**Fig. 3 | The performance characterization of QLEDs. a** EL spectra of ordered QD-based QLED under different driving voltages. Inset, QLED under operation (left) and schematic diagram of the QLED device structure (right). **b–d** Current density-luminance-voltage (*J-L-V*) characteristics, EQE-current efficiency-luminance characteristics and EQE statistical histogram of ordered and disordered QD-based QLEDs. **e** Luminance and time dependency characteristics curves of ordered and disordered QD-based QLEDs. **f** Surface temperature over time of ordered and disordered QD-based QLEDs under constant current density.

QLED with long-range ordered configuration decays to 95% of its initial luminance ($L_0 = 4868$ cd m$^{-2}$) after ~3 h, while the disordered configuration QLED decays to 95% of its initial luminance ($L_0 = 4781$ cd m$^{-2}$) after about 0.8 h. By fitting the equation $L_0^n T_{95} = $ constant[36], it is inferred that the QLED with long-range ordered configuration ($n = 1.8$, Supplementary Fig. 20a) has a $T_{95}$ of 54 h at 1000 cd m$^{-2}$, which is 3.6 times longer than the $T_{95}$ of 15 h for the disordered QLED ($n = 1.78$, Supplementary Fig. 20b). Additionally, $T_{50}$ operational lifetime measurements were also conducted (Supplementary Fig. 20c, d), the $T_{50}$ operational lifetimes are calculated to be $T_{50}$@1000 cd m$^{-2}$ = 390 h for ordered QD-based QLED and $T_{50}$@1000 cd m$^{-2}$ = 163 h for disordered QD-based QLED. The ordered QD-based QLED developed in this work is one of the most stable blue QLEDs (Supplementary Table 2). The order of the QDs also influences the heat dissipation of the QLED devices (Fig. 3f). After continuous operation for 10 h at room temperature, the surface temperature of the disordered QLED increases more rapidly, reaching 30.9 °C, which is 6.7 °C higher than the surface temperature of the QLED with long-range ordered configuration (24.2 °C). The EL intensity of the QLEDs is observed using confocal fluorescence microscopy (Supplementary Fig. 21), and the QLED equipped with ordered QD arrays exhibits a uniform EL intensity. To further understand the performance improvement of the blue QLED after introducing the microstructure, a 2D finite-difference time-domain (FDTD) simulation is used to model the light field distribution (further details can be found in "Methods"). The presence of the microstructure enables more light to escape from the device (Supplementary Fig. 22), which effectively enhances the device performance.

## High-resolution display device and static patterns
The long-range ordered blue QD microstructure arrays prepared using an aromatic-enhanced capillary bridge confinement strategy, when integrated into QLED devices, not only achieve high performance but also maintain high-resolution characteristics. We configured QLEDs with five different resolutions, featuring pixel sizes of 20 × 20 μm, 10 × 10 μm, 5 × 5 μm, 4 × 4 μm, 3 × 3 μm (Fig. 4a and Supplementary Fig. 23). Microscopic EL images reveal that blue QLEDs with different pixel sizes emit uniform and bright light. Statistical analysis of the EQE (Fig. 4b), maximum luminance (Fig. 4c), and operational lifetime (Fig. 4d) for these five different pixel size devices indicates that the blue QLED with a pixel size reduces to 3 μm has an average EQE of 18.3%, an average luminance of $7.3 \times 10^4$ cd m$^{-2}$, and an average $T_{95}$ lifetime ($L_0 = 1000$ cd m$^{-2}$) of 48 h. While maintaining high EQE and luminance, a resolution exceeding 5000 PPI is also achieved (see

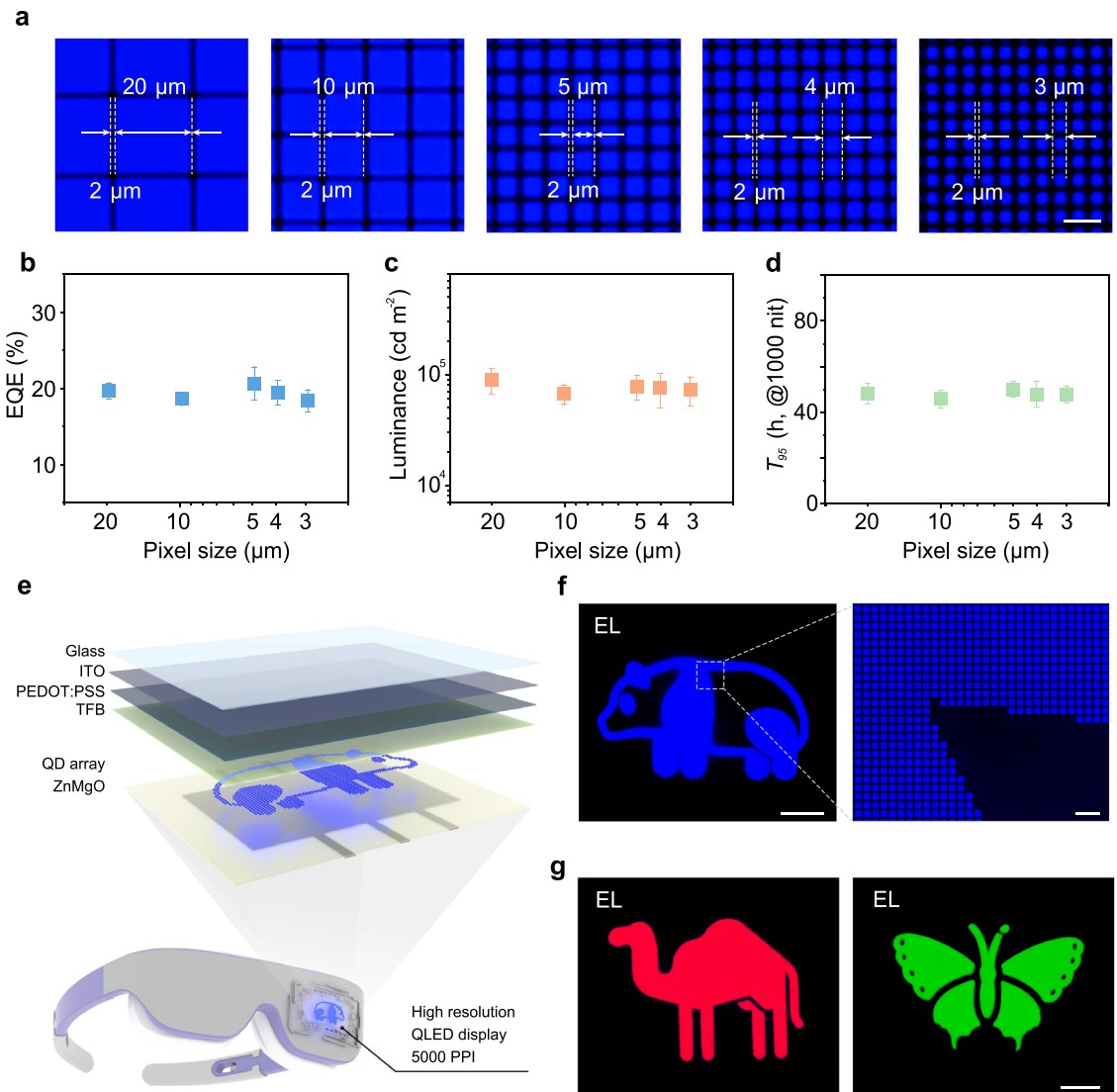

**Fig. 4 | Performance of high-resolution devices and static EL patterns. a** EL microscopy images of QLEDs with different pixel size (20 × 20 μm, 10 × 10 μm, 5 × 5 μm, 4 × 4 μm, 3 × 3 μm, scale bar is 10 μm). **b–d** Maximum EQE, maximum luminance, and operational lifetime of QLEDs with different pixel size. Error bars represent the standard deviation of five independent samples. **e** Schematic illustrating the structure of high-resolution QLED display. **f** Working state of high-resolution QLEDs (scale bar is 2 mm) and detailed images enlarged under a microscope (scale bar is 20 μm). **g** Static EL pattern display of red and green QLEDs (scale bar is 2 mm).

calculation formula in Supplementary Note 4), which has not been reported previously in high-resolution blue QLEDs. Finally, we demonstrate a blue high-resolution QLED display device with a size of 10 × 10 mm (Supplementary Fig. 24). The display device features a structure of ITO/PEDOT:PSS/TFB/QDs/ZnMgO/Al (Fig. 4e), notably, the QD layer is the long-range ordered blue QD microstructure patterned arrays prepared using the aromatic-enhanced capillary bridge confinement strategy. The high-resolution QLED display device successfully displays clear images (Fig. 4f), with complex curves. Further magnification is used to assess the uniformity of the pixels and the yield of high-quality pixels. Some visual defects are found in a 528-pixel extraction area, calculating a pixel yield of 99.6%. We have utilized this strategy to fabricate microstructured patterned arrays of red and green QDs. The static EL images of the red and green QLEDs exhibit uniform luminescence, as shown in Fig. 4g. The trichromatic QLED display devices with high pixel uniformity and high yield verify the feasibility and scalability of the aromatic-enhanced capillary bridge confinement strategy in display applications.

## Discussion

In summary, we develop an aromatic-enhanced capillary bridge confinement technique to achieve long-range order in blue QDs. By integrating these long-range ordered QD microstructure arrays into QLEDs, we successfully reduce the pixel size to a minimum of 3 μm and achieve a resolution exceeding 5000 PPI. This advancement allows for the display of EL static patterns with a pixel size of 5 μm. Moreover, the high-resolution QLEDs reach peak EQE of 24.1%, a maximum luminance of 101,519 cd m$^{-2}$, and exhibit high stability, with an extrapolated $T_{95}$ lifetime of 54 h at a luminance of 1000 cd m$^{-2}$. This work provides possibilities for the commercialization of QLEDs.

## Methods
### Materials

Cadmium oxide (CdO, 99.99%, powder), zinc acetate (Zn(OAc)$_2$, 99.99%), zinc oxide (ZnO, 99.9%, powder), magnesium acetate tetrahydrate (Mg(OAc)$_2$·4H$_2$O, 99%), oleic acid (OA, 90%) and sulfur (S, 99.998%, powder) were purchased from Sigma-Aldrich. 1-octadecene

(ODE, 90%) and selenium (Se, 99.999%, powder) were purchased from Thermo Scientific. Trioctylphosphine (TOP, 90%), dimethyl sulfoxide (DMSO, 99.7%), zinc acetate dihydrate (Zn(OAc)$_2$·2H$_2$O, 99.995%), tetramethylammonium hydroxide pentahydrate (TMAH·5H$_2$O, 97%) and heptadecafluorodecyltrimethoxysilane (FAS, 98%) were purchased from Aladdin. Cyclohexane, 3-fluorocinnamate (3-F-CA) and ethanol, n-octane, hexanes were obtained from Beijing Chemical Reagent Ltd, China. Poly(3,4-ethylenedioxythiophene):poly (styrenesulfonate) (PEDOT:PSS, AI 4083), photoresist, acetone and isopropanol were purchased from Beijing Innochem Co., Ltd. Poly(9,9-dioctylfluorene-co-N-(4-butylphenyl) diphenylamine) (TFB) was acquired from Volt-Amp Optoelectronics Tech. Co., Ltd.

## Synthesis of quantum dots

All precursor solutions were prepared under a N$_2$ atmosphere in a glovebox. S powder (10 mmol) was dissolved in 20 mL of TOP under continuous stirring until a homogeneous, clear solution was obtained with TOP-S solution concentration of 0.5 mmol mL$^{-1}$. Similarly, Se powder (10 mmol) was dissolved in 20 mL of TOP under continuous stirring until complete dissolution, yielding a transparent solution with TOP-Se solution concentration of 0.5 mmol mL$^{-1}$. A stoichiometric mixture of selenium and sulfur powders (5 mmol each) was dissolved in 20 mL of TOP under continuous stirring, resulting in a uniform and clear solution with TOP-Se/S solution concentration of 0.5 mmol mL$^{-1}$.

The synthesis was carried out by loading a mixture of CdO (0.07 mmol), Zn(OAc)$_2$ (2 mmol), OA (3 mL), and ODE (7 mL) into a 100 mL three-neck flask. The system was first degassed at 150 °C for 20 min and then heated to 300 °C under a constant nitrogen flow. Upon reaching the target temperature, a rapid injection of 1 mmol TOP-Se precursor solution was performed, and the reaction was allowed to proceed for 15 min. After cooling to room temperature, purify with hexane and ethanol to obtain 10 mL of QDs core solution. The purified core QDs were combined with 15 mL of Zn(OA)$_2$ solution in a 100 mL three-neck flask. The mixture was degassed for 20 min and subsequently heated to 300 °C under N$_2$. Using a syringe pump, a mixture of TOP-Se (1 mL), TOP-Se/S (1.5 mL), and TOP-S (2 mL) precursor solutions was introduced at a controlled rate of 5 mL h$^{-1}$, then react for 10 min. After the reaction is over, cool to room temperature and purify with hexane and ethanol.

## Ligand exchange of quantum dots

One microliter of QD stock solution was purified with n-hexane and ethanol and then dissolved in 1 mL of toluene. In an atmosphere of N$_2$, add 250 μL of a toluene solution of 3-F-CA with a concentration of 2.87 mM. The mixture was stirred in a N$_2$-filled glove box for 24 h and then purified again with ethanol.

## Synthesis of ZnMgO nanocrystals

A solution was prepared by dissolving Zn(OAc)$_2$·2H$_2$O (2.7 mmol) and Mg(OAc)$_2$·4H$_2$O (0.3 mmol) in 30 mL of DMSO. Separately, TMAH·5H$_2$O (0.5 mmol) was dissolved in 10 mL of anhydrous ethanol. The ethanolic TMAH solution (8.85 mL) was then gradually introduced into the precursor solution under continuous stirring for 2 h. The resulting mixture underwent purification through two cycles of washing with n-hexane and anhydrous ethanol. The final product was obtained by dispersing the purified precipitates in anhydrous ethanol to yield a ZnMgO nanoparticle solution with a concentration of 20 mg mL$^{-1}$.

## Fabrication of micropillar templates and modification

The micropillar template was prepared using photolithography and deep reactive ion etching (DRIE). A <100> oriented silicon wafer with a diameter of 10 cm was used. A direct laser writing system (Heidelberg DWL200) was employed to transfer the computer-predefined design with precision to ~1 μm onto the photoresist coated on the silicon

wafer, which was coated with the Shipley Microposit S1800 series. Subsequently, deep reactive ion etching with a fluorine-based reagent (Alcatel 601E) was performed to fabricate the periodic micropillar structures. After removing the photoresist (Microposit Remover 1165) and cleaning with ethanol and acetone, the micropillar template was generated.

We initially pressed a substrate coated with SU8 photoresist onto the tops of the micropillar. The tops of the micropillars were preserved by carefully removing the substrate. Subsequently, the sidewalls and gap regions of the micropillar topographical templates were rendered hydrophobic by exposure to FAS. For the FAS modification, the top-protected micropillar topographical templates were placed into a glass culture dish, followed by the addition of 20 μL of FAS liquid. The micropillar topographical templates were silanized with FAS molecules in a low-pressure environment at room temperature for 24 h, after which they were heated to 90 °C for 2 h. Following the modification with FAS, the SU-8 layer was removed using acetone to achieve asymmetric wettability on the tops and sidewalls of each micropillar.

## Fabrication of quantum dot arrays

The substrate was coated with a negative photoresist via spin-coating at 3000 rpm, followed by a soft-bake step at 90 °C for 1 min to remove residual solvents. Photolithographic patterning was performed using a custom-designed mask, followed by development in negative photoresist developer for 20 s, rinsing with negative photoresist rinse solution for 20 s, dried using nitrogen gas flow, and finally hard-baked at 150 °C for 5 min to create microporous structures with precisely controlled dimensions. Prior to QD deposition, the photoresist-patterned substrate was precisely aligned with a micropillar template using a high-precision alignment system. A 10 μL QD solution was carefully dispensed at the interface between the substrate and template. A custom-designed pressure application apparatus was employed to maintain optimal interfacial pressure and mechanical equilibrium throughout the solvent evaporation process. The assembled system was maintained at ambient temperature for 12 h to facilitate complete evaporation of the organic solvent phase. Following solvent dewetting and full evaporation, well-defined QD array patterns spontaneously formed on the substrate surface through self-assembly processes.

## Device fabrication

The optoelectronic devices were constructed on commercially available ITO-coated glass substrates, which underwent a sequential cleaning procedure involving ultrasonic treatment with detergent, deionized water, acetone, and isopropanol (15 min each), followed by UV-ozone surface activation for 15 min. PEDOT:PSS was deposited via spin-coating (4000 rpm, 40 s) and thermally annealed at 150 °C for 30 min under ambient conditions. Subsequently, TFB was applied by spin-coating an 8 mg mL$^{-1}$ chlorobenzene solution (3000 rpm, 40 s) and similarly baked at 150 °C for 30 min. Next, QD arrays were formed on the substrate according to the fabrication of QD arrays. A ZnMgO layer was then deposited via spin-coating (3000 rpm, 40 s) from solution and thermally cured at 60 °C for 30 min. Aluminum electrodes were thermally evaporated under ultrahigh vacuum conditions using standard deposition techniques. Finally, the completed devices were hermetically encapsulated with UV-curable epoxy resin and cover glass in a glovebox.

## Characterization of devices

Device characterization was performed under nitrogen atmosphere conditions using custom-designed test fixtures. Electroluminescent properties including current density-voltage-luminance (J-V-L) characteristics, EL spectra, and EQE were systematically evaluated using a commercial system (XPQY-EQE-Adv from Xipu Optoelectronics Technology Co., Ltd, Guangzhou). Lifetime tests were conducted on

encapsulated devices maintained in nitrogen environment throughout the testing duration.

## DFT calculations methodology

Theoretical calculations were performed within the framework of DFT using the generalized gradient approximation with the Perdew–Burke–Ernzerhof (PBE) functional. The electronic structure calculations employed the projector augmented-wave (PAW) method with a plane-wave basis set, as implemented in the VASP. Structural optimizations were conducted with a Monkhorst-Pack grid sampling of $1 \times 1 \times 1$ k-points in the Brillouin zone. The calculations utilized a plane-wave cutoff energy of 500 eV, with convergence criteria set to $1 \times 10^{-4}$ eV per atom for total energy and 0.02 eV $Å^{-1}$ for atomic forces.

## Optical simulation

The optical simulation was conducted using the finite-difference time-domain (FDTD) method. The optical parameters for each layer were actually measured using an ellipsometer. During the computation, the x-direction was set to periodic boundary conditions, while the y-direction was assigned perfectly matched layer conditions. Importantly, to achieve accurate computational results, the entire simulation area was divided into a grid of $2 \times 2$ nm. A dipole light source was employed as the excitation source. Finally, an electromagnetic field monitor was used to obtain the electric field distribution of the nanostructures.

## Characterizations

Fluorescent images were performed using an optical microscope from Vision Engineering Co. (UK). UV-visible absorption measurements were conducted with a Cary 7000 spectrophotometer (Agilent, USA). PL properties, including emission spectra, PLQY, and transient PL decay, were characterized using an FLS 1000 spectrometer (Edinburgh Instruments). The refractive index was obtained using an ellipsometer (Smart SE). Surface morphology was examined by AFM (Bruker Nano Inc.). SEM images were acquired at 10 kV and 10 μA using a Hitachi SU8010 system (Japan). TEM (JEOL 2100 F, Japan) operating at 200 kV was employed to determine QD size and morphology, with lattice structure analysis performed using a double spherical aberration-corrected TEM (JEM-ARM300F, Japan) at 300 kV. XRD patterns were collected on a Bruker diffractometer operating at 40 kV and 40 mA. The QD arrangement was investigated by GISAXS (Xeuss 2.0, France) with Cu Kα radiation ($\lambda = 1.54$ Å) and a Pilatus3R detector at 0.3° incidence angle. DLS measurements were carried out using a Dybapro NanoStar instrument (OPTILAB Rex, HELEOSII). $^1$H NMR spectra were recorded on an Ascend 600 spectrometer (Bruker Nano Inc.) with samples dissolved in deuterated toluene. Surface chemical composition was analyzed by XPS using an ESCALAB 250Xi system.

## Data availability

All data supporting the findings of this study are available within the paper and its Supplementary files. Any additional information related to the study is available from the corresponding author upon request. Source data are provided with this paper.

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

## Acknowledgements

This work was supported by the National Natural Science Foundation of China (T2425026, 52173190 and 21988102 to Y.C.W., 22122206 to Z.Y.H., 22205077 to H.F.G., and 52403310 to Y.Y.Z.), the National Natural Sci-ence Foundation of China Joint Fund for Regional Innovation and Development (U24A20495 to Z.Y.H.) and Youth Innovation Promotion Association CAS (2018034 to Y.C.W.).

## Author contributions

Y.J., H.L., N.G., and T.L. fabricated the devices and collected the per-formance data of the QLEDs. Y.J. and H.L. synthesized the materials. Y.J. and H.L. wrote the manuscript. Y.J., H.L., F.L., H.M., Y.Z., D.W., and J.F. conducted data analysis. Y.Z., H.G., Z.H., and Y.W. provided financial support. H.L., Z.H., L.J,. and Y.W. directed the project. All authors con-tributed to the scientific discussion and modified the manuscript.

## Competing interests

The authors declare no competing interests.
