## [Transparent Peer Review file · Nature Communications]

Long-Range Order Enhance Performance of Patterned Blue Quantum Dot Light-Emitting Diodes

Corresponding Author: Professor Yuchen Wu

Version 0:

Reviewer comments:

Reviewer #1

(Remarks to the Author)

Jia et al. reported that they have developed a strategy for assembling a long-range ordered array of blue QD microstructures using an aromatic-enhanced capillary bridge confinement. When integrating these ordered QD microstructure arrays into QLEDs, they observed much improved QLED performance. Considering that the device performance was achieved in a microscale array with a resolution density of up to 5000 PPI, the EQE and operational lifetime are very impressive. This study effectively addresses the long-standing challenges of low efficiency, poor stability, and scalability associated with patterned QLEDs, marking a breakthrough in the development of patterned QLEDs. Overall, this work is highly innovative and showcases the vast potential of QD display technology. If the following issues/suggestions are adequately addressed, I recommend that this work be published.

1. Many reports suggest that reducing the distance between particles and decreasing the disorder in the assembly enhance Förster resonance energy transfer (FRET) between QDs (ACS Energy Letters 2017, 2(1), 154-160), which in turn leads to poorer exciton confinement and stronger non-radiative recombination, thereby reducing device performance. This appears to be in contrast to the mechanism proposed in this work.
2. To determine the long-range ordered arrangement in the QD films, it cannot be conclusively confirmed whether processing with QD microstructures and micropillar templates leads to a more compact structure solely based on the author's grazing-incidence small-angle X-ray scattering (GISAXS) characterization. To assess whether the large-area films exhibit ordering, I suggest using field-effect transistor (FET) tests to compare the film conductivity.
3. The authors state that QLEDs with long-range ordered structures exhibit lower current density before turn-on, which is attributed to reduced charge leakage. To enhance the credibility of this conclusion, it is recommended that the authors provide additional data to support this claim.
4. The concept of "capillary bridge" is popularly used for guiding the formation of microstructures from liquid materials, and it is probably recognizable as a new way for blue colloidal QDs. However, spending over 12 hours on a single step in the fabrication process is not very appealing to LED fabrication. Is there a way to significantly speed up the process without compromising the device's performance.
5. The manuscript mentions that a microhole array substrate was fabricated using a photolithography process to achieve patterning, but the key process parameters are not sufficiently detailed. To ensure the reproducibility of the experimental results, it is recommended that the authors provide additional information on the relevant photolithography parameters (such as development time and exposure time).

Reviewer #2

(Remarks to the Author)

The manuscript investigates the application of 3-fluorocinnamate (3-F-CA) ligand exchange in CdSe-based core/shell QDs and its effect on achieving a long-range ordered configuration via a capillary bridge confinement strategy. While the study builds upon a previously reported capillary-bridge-assisted QD assembly approach (Adv. Mater. 2022, doi:10.1002/adma.202110695), which limits the novelty of the patterning method itself, the reported device performance is impressive. This work introduces 3-F-CA ligands as a means to promote QD ordering and enhance optoelectronic performance. Although the performance improvement is compelling, several key issues must be addressed to substantiate

the central claims, particularly those related to the role and mechanism of the 3-F-CA ligands.

1. The manuscript emphasizes the π - π interaction between 3-F-CA ligands as a novelty, yet there is little experimental discussion supporting this mechanism except the DFT calculation. In particular, the DFT simulations likely assume complete ligand substitution with 3-F-CA, whereas Supporting Figure 7b clearly indicates only partial exchange. Given the presence of residual oleate ligands, it is plausible that steric hindrance significantly limits the effective π - π stacking between aromatic groups. The authors should elaborate on the implications of incomplete ligand exchange and discuss how π - π interactions may be influenced under realistic surface coverage conditions.

Additionally, direct experimental validation of π - π interactions between 3-F-CA ligands is necessary to support the theoretical claims. If spectroscopic evidence (e.g., UV-Vis, FTIR, solid-state NMR) or structural characterization (e.g., grazing incidence XRD) is inconclusive, a comparative study using a non-aromatic ligand with a similar chain length might be an alternative way to investigate the effect of π - π interactions.

2. The authors propose that excess 3-F-CA ($> 0.7 \mu\text{mol}$) leads to QD surface degradation and attributed it to the increased attraction between QDs and consequent QD aggregation. However, the correlation between PLQY loss and aggregation has not been clearly established. It is speculated that the PLQY loss is due to the change in the density of bound ligands or insufficient passivation capability of 3-F-CA-dominated ligand system. Additional chemical and photophysical characterizations are necessary to investigate the cause of the observed optical degradation.

3. Most experiments are conducted using QDs modified with $0.7 \mu\text{mol}$ of 3-F-CA. Under these conditions, can the extent of ligand exchange be quantified? Quantitative NMR, XPS, or TGA analysis would help determine the ligand density and offer insight into how this correlates with the observed assembly behavior and device performance.

4. Supporting Figure 7b shows a sharp NMR signal at 5.4 ppm, likely corresponding to free oleic acid. This is unexpected given that a purification step was performed following ligand exchange. The authors should clarify the origin of this signal—whether it reflects incomplete purification, equilibrium between bound and free ligands, or re-adsorption of oleic acid during sample handling.

5. The authors attribute the improved device performance to enhanced QD ordering induced by the capillary bridge method. While surface roughness is mentioned, additional structural analyses—such as AFM for measuring line edge roughness or taper angle—would provide a more comprehensive view of the pixel morphology. Including these data would reinforce the manuscript.

6. In the SCLC analysis, the authors appear to assume a trap-free model to extract charge carrier mobility. However, this assumption may oversimplify the actual transport behavior, as trap states are commonly present in QD-based systems. Incorporating a model that considers either a single trap level or a distribution of trap states (e.g., Gaussian disorder model) may yield a more accurate assessment of the mobility. Furthermore, the authors are encouraged to discuss whether hole injection efficiency can be estimated from the SCLC or dark-injection SCLC (DI-SCLC) measurements.

7. The manuscript reports the device operational stability in terms of T95 at 1000 nit. It would be beneficial to include additional lifetime metrics, such as T50, to provide a more comprehensive evaluation of device stability. Reporting multiple lifetime metrics not only enhances the credibility of the presented data but also facilitates more accurate comparisons with previously reported studies. In addition, given the remarkably high EQE values, independent validation through cross-laboratory measurements would further strengthen the reliability and reproducibility of the results.

Reviewer #3

(Remarks to the Author)

This manuscript presents a significant advancement in the field of blue quantum dot light-emitting diodes (QLEDs) by introducing an aromatic-enhanced capillary bridge confinement strategy to achieve long-range ordered blue QD arrays. This strategy was enabled by employing a ligand 3-C-FA, which introduces strong ligand-ligand interaction between small blue QDs that originally exhibit weak van der Waals interactions due to its small size/volume. By exploiting the long-range ordered QD arrays, the authors demonstrate excellent device performance with a peak external quantum efficiency (EQE) of 24.1%, a peak luminance of $101,519 \text{ cd m}^{-2}$, and an impressive T95 lifetime of 54 hours at $1,000 \text{ cd m}^{-2}$. Additionally, the authors achieve a minimum pixel size of $3 \mu\text{m}$, which corresponds to a potential resolution exceeding 5,000 pixels per inch (PPI). The research is well-structured, and the experimental methodology is well-articulated. However, the following concerns must be adequately addressed if its publication to Nature Communications is to be considered.

Major comments

Distinguishing the Role of Ligand Shortening vs. Ordering in Performance Enhancement

The key contribution of this work is that the enhanced ligand-ligand interaction improves the QD film ordering, leading to significant device performance improvements. However, an important factor to consider is that the introduction of 3-F-CA shortens the ligand length, reducing interparticle distance. It is crucial to distinguish whether the observed EQE enhancement arises predominantly from improved ordering or simply from decreased interparticle distance. A simple experiment to verify this would be to compare GISAXS measurements of randomly coated (e.g., spin-coated) films using QDs with 3-F-CA ligands (the film that yielded EQE of 16.4% in line 151. If interparticle distance is reduced but the ordering remains disordered, and if these disordered films show a significant performance difference from structured films, it would confirm that ordering plays the primary role. Another approach would be to test a ligand with a length similar to OA but with strong π - π interactions similar to 3-F-CA.

Applicability to Green and Red QLEDs: The authors focus exclusively on blue QDs, explaining that blue QDs have inherently weaker van der Waals forces due to their smaller size. However, it is unclear whether the same approach could be extended to green and red QDs. Does the introduction of 3-F-CA have marginal effect/weak effect on green/red QDs simply because their van der Waals interactions are already strong enough to induce ordering? Could 3-F-CA still contribute by further reducing interparticle distance in green/red QDs, thereby enhancing performance? Experimental verification using green and red QDs would clarify whether this strategy is beneficial beyond blue QDs.

Quantifying Interaction Strength for Long-Range Ordering

The authors perform DFT calculations to estimate ligand interaction strength, but it remains unclear how much interaction strength is required to achieve long-range ordering. Could the authors provide a quantitative threshold for interaction energy required for long-range ordering?

How do the interaction energies of green and red QDs compare, and do they naturally meet this threshold even without introducing 3-F-CA?

QD Size Considerations in Other Systems

The manuscript states that blue QDs used here have a diameter of ~8 nm. However, commonly used green/red QDs (e.g., InP) have similar diameters. Would applying 3-F-CA to InP QDs (or other large QDs) also lead to enhanced performance?

Surface Binding Energy Differences Between OA and 3-F-CA

The authors present binding energy calculations showing a significant difference between OA and 3-F-CA, despite both ligands binding via carboxylate groups. What accounts for this large discrepancy in binding energy? Does π - π stacking contribute directly to surface binding stability, or is another mechanism at play?

Quantification of Ligand Exchange and Interaction Strength Modulation The authors quantify OA reduction by 39% upon ligand exchange. However, there is no quantitative assessment of 3-F-CA incorporation. Can the amount of 3-F-CA on the surface be measured? Would further increasing 3-F-CA concentration enhance interparticle interactions and improve ordering?

Line 174, the manuscript reports differing results at low vs. high ligand concentrations. A more detailed explanation of why these variations occur would help improve clarity.

The manuscript reports a surprising 7°C temperature difference based on QD ordering, which is surprisingly high (although I do not have much experience on such measurement). If ordering significantly alters thermal conductivity to that extent, can you measure the thermal conductivity of these films and directly confirm if QD films simply changing the particle ordering can yield such a dramatic change in thermal conductivity? having different ordering?

Verification of PPI Calculation for 3 μ m Patterns The manuscript claims a resolution exceeding 5,000 PPI for 3 μ m pixel sizes. However, assuming an RGB subpixel structure, a full pixel would typically be larger than 3 μ m (likely ~10 μ m per pixel – 3 μ m blue + 3 μ m green + 3 μ m red). How was the 5,000 PPI figure calculated? I understand that this calculation depends on the design of the subpixels as well. It would be nicer to explain what assumptions were made regarding pixel layout.

Definition of Active Area in Device Performance Measurements The manuscript suggests that photoresist was used as a pixel-defining layer. How was the active area defined for current density and luminance calculations?

Version 1:

Reviewer comments:

Reviewer #1

(Remarks to the Author)

The authors have properly addressed my comments, and I believe it is now suitable for publication.

Reviewer #2

(Remarks to the Author)

As the authors sufficiently revised their manuscript to comply with the comments, I recommend publication of this manuscript without further revision.

Reviewer #3

(Remarks to the Author)

I consider that the authors have sufficiently addressed the comments/concerns raised by the three reviewers. I suggest its publication to Nature Communications.

Response to Reviewers' Comments and Revised Details

Manuscript ID: NCOMMS-25-10739

Response to Reviewer 1:

Comment 1: Jia et al. reported that they have developed a strategy for assembling a long-range ordered array of blue QD microstructures using an aromatic-enhanced capillary bridge confinement. When integrating these ordered QD microstructure arrays into QLEDs, they observed much improved QLED performance. Considering that the device performance was achieved in a microscale array with a resolution density of up to 5000 PPI, the EQE and operational lifetime are very impressive. This study effectively addresses the long-standing challenges of low efficiency, poor stability, and scalability associated with patterned QLEDs, marking a breakthrough in the development of patterned QLEDs. Overall, this work is highly innovative and showcases the vast potential of QD display technology. If the following issues/suggestions are adequately addressed, I recommend that this work be published.

Response to comment 1: We sincerely thank the Reviewer 1 for the positive evaluation and encouraging comments regarding our work. We are very pleased that the reviewer recognized the innovation and significance of our strategy for assembling long-range ordered blue QD microstructures via aromatic-enhanced capillary bridge confinement. As noted, the achievement of high EQE and extended operational lifetime in microscale patterned QLEDs with a resolution density up to 5,000 PPI demonstrates the potential of this method to address critical challenges in patterned QLEDs, including low efficiency, poor stability, and scalability. We greatly appreciate the Reviewer 1's recognition of our efforts and the broader impact this work may have on advancing QD display technologies. Following the Reviewer 1's suggestions, we have carefully revised the manuscript to address the remaining issues, as detailed below.

Comment 2: Many reports suggest that reducing the distance between particles and decreasing the disorder in the assembly enhance Förster resonance energy transfer (FRET) between QDs (ACS Energy Letters 2017, 2(1), 154-160), which in turn leads to poorer exciton confinement and stronger non-radiative recombination, thereby reducing device performance. This appears to be in contrast to the mechanism proposed in this work.

Response to comment 2: We sincerely appreciate the constructive comments from Reviewer 1. Indeed, reducing interparticle distance and minimizing disorder in assemblies could enhance Förster resonance energy transfer (FRET) between quantum

dots (QDs) (*Chem. Rev.* **116**, 10513 (2016)). However, disordered QD films may induce increased carrier loss and augmented diffusion during assembly, thereby adversely affecting radiative recombination processes. By optimizing the arrangement of QDs, carrier transport losses can be effectively mitigated, which proves crucial for enhancing radiative recombination efficiency (*Nature* **629**, 586 (2024); *eScience* **4**, 100227 (2024)). Accordingly, we evaluated the FRET effect by calculating the Förster radius (R_0) of ligand-exchanged blue QDs to systematically assess energy transfer characteristics (*Science* **359**, 6373 (2018); *J. Am. Chem. Soc.* **126**, 301 (2004); *Chin. Phys. B* **30**, 127802 (2021); *Adv. Opt. Mater.* **11**, 2202451 (2023)).

$$R_0 = 0.211 [\kappa^2 n^4 Q_D J]^{1/6}$$

where κ^2 represents the dipole orientation parameter, and n denotes the refractive index of the medium. Q_D is the PL QY of the donor. J is the integral of spectral overlap between the excitation spectrum of the donor and the absorption spectrum of the acceptor, calculated as follows:

$$J = \int f_D(\lambda) \varepsilon_A(\lambda) \lambda^4 d\lambda$$

where $f_D(\lambda)$ is the normalized emission spectrum of the donor, and $\varepsilon_A(\lambda)$ is the molar extinction coefficient of the acceptor. The surface-to-surface distance between ligand-exchanged QDs (5.2 nm) exceeds the calculated Förster radius (4.3 nm), indicating that FRET exerts negligible influence on both the excited-state lifetime and radiative recombination efficiency of the implemented QD system.

Comment 3: To determine the long-range ordered arrangement in the QD films, it cannot be conclusively confirmed whether processing with QD microstructures and micropillar templates leads to a more compact structure solely based on the author's grazing-incidence small-angle X-ray scattering (GISAXS) characterization. To assess whether the large-area films exhibit ordering, I suggest using field-effect transistor (FET) tests to compare the film conductivity.

Response to comment 3: We sincerely appreciate the constructive comments from Reviewer 1. In the revised manuscript, we fabricated field-effect transistors (FET) to determine the conductivity of the QD films (**Fig. R1**). The channel length (L) and width (W) of the transistors were 2 μm and 40 μm , respectively. Both films exhibited linear current-voltage (I - V) characteristics in the scanning voltage range from -2 V to 2 V. The conductivity (σ) of the two QD films was calculated using the following equation (*Nature* **629**, 586–591 (2024)):

$$\sigma = \frac{I}{V} \times \frac{L}{T \times W}$$

where L is the channel length, W is the channel width, T is the thickness of the QD film, and I/V is the slope of the current-voltage curve. The calculated conductivity of the ordered QD layer was found to be $4.2 \times 10^{-4} \text{ S m}^{-1}$, 3.2-fold higher than that of the disordered QD layer.

Fig. R1. Field-effect transistors of ordered and disordered QD films.

To enhance the quality of our manuscript, we have added relevant discussions in the revised version, as detailed below:

Page 8, Lines 6-9: We fabricated field-effect transistors (FET) to determine the conductivity of the QD films (Supplementary Fig. 15d)³⁵. The calculated conductivity of the ordered QD layer was found to be $4.2 \times 10^{-4} \text{ S m}^{-1}$ (Supplementary Note 3).

Comment 4: The authors state that QLEDs with long-range ordered structures exhibit lower current density before turn-on, which is attributed to reduced charge leakage. To enhance the credibility of this conclusion, it is recommended that the authors provide additional data to support this claim.

Response to comment 4: We appreciate the Reviewer 1's insightful comment. To substantiate our conclusion, we conducted capacitance-voltage ($C-V$) measurements (**Fig. R2**), which reveal that devices with long-range ordered QD layers exhibit a lower peak capacitance compared to control devices. This result indicates enhanced hole injection, reduced electron accumulation, and suppressed charge leakage, thereby supporting the observed lower current density before turn-on. Relevant data and discussion have been incorporated into the revised manuscript.

Fig. R2. The capacitance-voltage characteristics of ordered and disordered QD-based QLEDs.

To enhance the quality of our manuscript, we have added relevant discussions in the revised version, as detailed below:

Page 8, Lines 18-21: The current density-luminance-voltage characteristics of both QLEDs (Fig. 3b) show that the QLED with long-range ordered configuration has a lower current density before turn-on, indicating less charge leakage (Supplementary Fig. 17)³⁵.

Comment 5: The concept of "capillary bridge" is popularly used for guiding the formation of microstructures from liquid materials, and it is probably recognizable as a new way for blue colloidal QDs. However, spending over 12 hours on a single step in the fabrication process is not very appealing to LED fabrication. Is there a way to significantly speed up the process without compromising the device's performance.

Response to comment 5: We are very grateful for the detailed comments from Reviewer 1. Under the confinement of the capillary bridge, the self-assembly of nanoparticles follows the principle of minimizing Gibbs free energy, spontaneously forming stable, dense, and ordered QD microstructures. This requires that the self-assembly process of QDs be completed in a relatively slow manner (compared to spin coating), allowing sufficient time for QDs to undergo Brownian motion for uniform mass transfer and adequate self-assembly time. Therefore, to achieve long-range order of QDs, spending a longer time is necessary. We have also attempted to accelerate this process by measures such as increasing the temperature and blowing air to speed up solvent evaporation. However, rapid solvent evaporation at the gas-liquid interface causes severe Marangoni flow, which is detrimental to achieving long-range ordered assembly. The temperature difference at the gas-liquid interface is controlled within a narrow

range in the thermostatic chamber to avoid strong Marangoni flow, which could potentially accelerate this process (*eScience* 4, 100227 (2024)).

Comment 6: The manuscript mentions that a microhole array substrate was fabricated using a photolithography process to achieve patterning, but the key process parameters are not sufficiently detailed. To ensure the reproducibility of the experimental results, it is recommended that the authors provide additional information on the relevant photolithography parameters (such as development time and exposure time).

Response to comment 6: We sincerely appreciate the meticulous comments from Reviewer 1. In the revised manuscript, we have expanded the methods section to include comprehensive details. The specific modifications are as follows:

Page 16, Lines 7-17: A negative photoresist was spin-coated at 3,000 rpm and soft-baked at 90 °C for 1 min. The photoresist was exposed through a custom mask using a photolithography system, followed by development in negative photoresist developer for 20 s, rinsing with negative photoresist rinse solution for 20 s, drying with a nitrogen gun, and hard-baking at 150 °C for 5 min to create micropores with specific dimensions. An alignment system was employed to align the photoresist-patterned substrate with a micropillar template. A 10 µL QD solution was dispensed between the substrate and the micropillar template, and a homemade pressure device was utilized to maintain appropriate pressure and balance. The assembled system was left at room temperature for 12 h to allow evaporation of the organic solvent. After the dewetting and the total evaporation of liquids, QD array patterns formed on the substrate.

Response to Reviewer 2:

Comment 1: The manuscript investigates the application of 3-fluorocinnamate (3-F-CA) ligand exchange in CdSe-based core/shell QDs and its effect on achieving a long-range ordered configuration via a capillary bridge confinement strategy. While the study builds upon a previously reported capillary-bridge-assisted QD assembly approach (Adv. Mater. 2022, doi:10.1002/adma.202110695), which limits the novelty of the patterning method itself, the reported device performance is impressive. This work introduces 3-F-CA ligands as a means to promote QD ordering and enhance optoelectronic performance. Although the performance improvement is compelling, several key issues must be addressed to substantiate the central claims, particularly those related to the role and mechanism of the 3-F-CA ligands.

Response to comment 1: We sincerely thank Reviewer 2 for their valuable time, effort, and constructive comments on our manuscript. We acknowledge that the capillary-bridge-assisted self-assembly strategy employed in this study builds upon our group's previous work. However, the primary objective of the current study is to address the longstanding performance bottlenecks in patterned QLEDs, where non-uniform deposition and morphological disruption of the emissive layer at the microscale are key factors limiting device efficiency and stability. The innovation of this work lies in the introduction of functional ligands, which enhance the coupling between small-sized QDs through intermolecular interactions, thereby effectively suppressing the adverse effects of complex fluid dynamics on the assembly process and enabling long-range ordering of the QDs. Specifically, the incorporation of 3-fluorocinnamate (3-F-CA) as a ligand significantly enhances the assembly driving force of blue QDs, leading to the formation of highly ordered, dense, and uniform microstructured arrays under confined conditions. The resulting optimization of the emissive layer morphology markedly improves charge transport balance and radiative recombination efficiency in the devices, ultimately achieving simultaneous enhancement in performance and operational stability. Therefore, we believe this work offers a universal and scalable strategy for achieving high-performance patterned blue QLEDs and holds significant implications for advancing the practical application of QD display technologies. We are confident that this study possesses sufficient novelty and technical depth to merit publication in *Nature Communications*.

Comment 2: The manuscript emphasizes the π - π interaction between 3-F-CA ligands as a novelty, yet there is little experimental discussion supporting this mechanism except the DFT calculation. In particular, the DFT simulations likely assume complete ligand substitution with 3-F-CA, whereas Supporting Figure 7b clearly indicates only

partial exchange. Given the presence of residual oleate ligands, it is plausible that steric hindrance significantly limits the effective π - π stacking between aromatic groups. The authors should elaborate on the implications of incomplete ligand exchange and discuss how π - π interactions may be influenced under realistic surface coverage conditions.

Additionally, direct experimental validation of π - π interactions between 3-F-CA ligands is necessary to support the theoretical claims. If spectroscopic evidence (e.g., UV-Vis, FTIR, solid-state NMR) or structural characterization (e.g., grazing incidence XRD) is inconclusive, a comparative study using a non-aromatic ligand with a similar chain length might be an alternative way to investigate the effect of π - π interactions.

Response to comment 2: We sincerely thank Reviewer 2 for the thoughtful critique and constructive suggestions. In this study, the strategy of partially retaining the original long-chain oleic acid ligands was adopted to balance the synergistic regulation requirements for colloidal stability and ordered assembly of QDs. Although complete replacement of long-chain ligands might enhance the π - π interactions between 3-F-CA ligands, it would lead to a drastic reduction in surface steric hindrance of QDs, inducing destabilization and aggregation of the colloidal system (*Adv. Sci.* **8**, 2101125 (2021); *Nature* **586**, 385-389 (2020); *Nature* **575**, 634-638 (2019)). In contrast, the moderate introduction of 3-F-CA ligands precisely enhanced inter-QD interactions without compromising colloidal stability.

We also replaced the OA ligands on the QD surface with 3-Cyclohexylpropionic acid (3-CPA, it has a chain length comparable to that of 3-F-CA). The capillary bridge-confined assembly strategy was employed to fabricate two types of QD microstructure arrays. The 3-CPA modified QDs exhibited disordered arrangement (**Fig. R4a**), while the 3-F-CA modified QDs demonstrated long-range ordered alignment (**Fig. R4b**). Furthermore, these two QD microstructure arrays were integrated into QLED devices, and their external quantum efficiency (EQE)-luminance curves as well as EQE distribution histograms are shown in **Figs. R4c** and **d**. The 3-CPA-modified QD-based QLED devices achieved a maximum EQE of 15.4%, a peak luminance of 45,023 cd m⁻², and an average EQE of 14.6%. In contrast, the 3-F-CA-modified QD-based QLED devices exhibited a maximum EQE of 24.1%, a peak luminance of 101,519 cd m⁻², and an average EQE of 22.0%. These results indicate that the presence of π - π interactions significantly influences the long-range ordered assembly of QDs.

Fig. R4 SEM image of **a**, 3-F-CA and **b**, 3-CPA modified QDs. (scale bar is 20 nm). **c**, EQE- luminance curves and **d**, EQE statistical histogram of assembled 3-CPA and 3-F-CA modified QD-based QLEDs.

To enhance the quality of our manuscript, we have added relevant discussions in the revised version, as detailed below:

Page 7, Lines 25-29: We also replaced the OA ligands on the QD surface with 3-Cyclohexylpropionic acid (3-CPA). Notably, 3-CPA modified QDs fabricated via the capillary bridge-mediated confined assembly strategy exhibited disordered packing (Supplementary Fig. 14), further demonstrating that π - π interactions promote long-range ordered packing of QDs.

Comment 3: The authors propose that excess 3-F-CA ($> 0.7 \mu\text{mol}$) leads to QD surface degradation and attributed it to the increased attraction between QDs and consequent QD aggregation. However, the correlation between PL QY loss and aggregation has not been clearly established. It is speculated that the PL QY loss is due to the change in the density of bound ligands or insufficient passivation capability of 3-F-CA-dominated ligand system. Additional chemical and photophysical characterizations are necessary to investigate the cause of the observed optical degradation.

Response to comment 3: We sincerely appreciate Reviewer 2's constructive suggestions. During the ligand exchange process, significant flocculation and turbidity

occurred upon reaching a specific addition amount (*Nat. Energy* **10**, 592 – 604 (2025); *Nat. Commun.* **11**, 103 (2020)). Consequently, operability was compromised during testing. To ensure more rigorous data presentation, we modified the figure in the manuscript.

Fig. R5 a DLS volume-based size distribution graphs of QDs ligand exchange with the different concentration of the 3-F-CA (0 μmol, 0.1 μmol, 0.4 μmol, 0.7 μmol). **b** The PL QY of the QD solution after addition of varying amounts of 3-F-CA (0 μmol, 0.1 μmol, 0.4 μmol, 0.7 μmol). Top left inset, photograph taken under illumination at 365 nm; from left to right: 0 μmol, 0.1 μmol, 0.4 μmol, 0.7 μmol. **c** The transient PL decay of the QD solution after addition of varying amounts of 3-F-CA (0 μmol, 0.1 μmol, 0.4 μmol, 0.7 μmol).

To enhance the quality of our manuscript, we have added relevant discussions in the revised version, as detailed below:

Page 5, Lines 24-33: To determine the optimal amount of 3-F-CA, we compared the optical properties of QDs with different amounts of 3-F-CA. As the amount of 3-F-CA increased, the PL QY of the QDs gradually increased. At an addition amount of 0.7 μmol, the PL QY of the QD solution increased to 93% (Fig. 1f). The transient PL decay maintained a single-exponential profile, with a value increased to 16.9 ns, which is higher than that of OA-modified QD solution (12.1 ns, Fig. 1g). This is likely due to the stronger binding ability of 3-F-CA to the QD surface, which can more effectively passivate the surface²⁸. The dynamic light scattering (DLS) diameter of the QD solution gradually increases with the addition of 3-F-CA (Fig. 1e), further demonstrating the impact of 3-F-CA on the interactions between the QDs.

Comment 4: Most experiments are conducted using QDs modified with 0.7 μmol of 3-F-CA. Under these conditions, can the extent of ligand exchange be quantified? Quantitative NMR, XPS, or TGA analysis would help determine the ligand density and offer insight into how this correlates with the observed assembly behavior and device

performance.

Response to comment 4: We sincerely appreciate the reviewer 2's insightful suggestion. To determine the ligand density of 3-F-CA on the QDs surfaces, deconvolution analysis of the X-ray photoelectron spectroscopy (XPS) peaks was performed, as shown in **Fig. R6** (*Nature* **586**, 385-389 (2020); *Adv. Funct. Mater.* **2501770**, (2025), <https://doi.org/10.1002/adfm.202501770>). The Zn 2*p* orbital signals originated from the outermost ZnS layer of the QDs (orange peaks) and Zn bound to carboxylic acid (-COOH) groups (green peaks). The integrated peak area of F was utilized to quantify the relative number of 3-F-CA ligands. The Zn and F peaks were integrated, and the corresponding atomic ratio was calculated by normalizing with their XPS sensitivity factors. The atomic ratio was determined using the following formula:

$$R_{X:Y} = \frac{\frac{A_X}{S_X}}{\frac{A_Y}{S_Y}}$$

where A_X and A_Y are the XPS peak areas of elements X and Y obtained from spectral integration, and S_X and S_Y are the sensitivity factors for elements X and Y. Using the Zn signal associated with carboxylic acid (-COOH) groups in the QDs as a reference (**Table R1**), the normalized ratio of F was determined to be 0.84.

Fig. R6 a, XPS spectra of Zn 2*p* orbit of the 3-F-CA modified QD film. **b**, XPS spectra of F 1*s* orbit of the 3-F-CA modified QD film.

Table R1. Atomic ratios of surface elements in the 3-F-CA modified QD film derived from XPS results.

Element	Area	RSF	Atomic ratio
Zn (Zn-O)	46,669	28.72	1
F	6,044	4.43	0.84

To enhance the quality of our manuscript, we have added relevant discussions in the revised version, as detailed below:

Page 5, Lines 33-36; Page 6, Lines 1: To determine the ligand density of 3-F-CA on the QDs surfaces, deconvolution analysis of the X-ray photoelectron spectroscopy (XPS) peaks was performed (Supplementary Fig. 7). Using the Zn signal associated with carboxylic acid (-COOH) groups in the QDs as a reference (Supplementary Table 1), the normalized ratio of F was determined to be 0.84.

Comment 5: Supporting Figure 7b shows a sharp NMR signal at 5.4 ppm, likely corresponding to free oleic acid. This is unexpected given that a purification step was performed following ligand exchange. The authors should clarify the origin of this signal—whether it reflects incomplete purification, equilibrium between bound and free ligands, or re-adsorption of oleic acid during sample handling.

Response to comment 5: We sincerely appreciate Reviewer 2's insightful comments. After ligand exchange in deuterated toluene without purification, free OA signals were observed, indicating successful ligand replacement. We sincerely apologize for the lack of clarity in describing this process due to our oversight. Therefore, we revised the experimental protocol and provided a more detailed description. Specifically, after 3-F-CA ligand exchange, purification was performed using ethanol and deuterated toluene, followed by redispersing the QDs in deuterated toluene for ^1H NMR characterization (**Fig. R7**). Both QD samples exhibited characteristic proton signals of coordinated oleic acid ligands, and the signals from free oleic acid completely disappeared. Notably, specific proton signals of 3-F-CA were only observed in the 3-F-CA modified QDs, confirming the success of the ligand exchange.

Fig. R7 ^1H NMR spectra of **a**, OA and **b**, 3-F-CA modified QDs.

To enhance the quality of our manuscript, we have added relevant discussions in the revised version, as detailed below:

Page 5, Lines 12-14: Hydrogen signals from 3-F-CA were exclusively observed in the ^1H nuclear magnetic resonance (NMR) spectra of 3-F-CA-modified QDs (Supplementary Fig. 4).

Comment 6: The authors attribute the improved device performance to enhanced QD ordering induced by the capillary bridge method. While surface roughness is mentioned, additional structural analyses—such as AFM for measuring line edge roughness or taper angle—would provide a more comprehensive view of the pixel morphology. Including these data would reinforce the manuscript.

Response to comment 6: We sincerely appreciate the reviewer 2's constructive suggestion. We analyzed the insulating photoresist layer prior to QD deposition with AFM (**Fig. R8a**). Height profiles along the blue and orange solid lines (**Fig. R8b**) revealed well-defined square microhole arrays with a depth of ~ 30 nm. After the deposition of 3-F-CA-modified QDs assembled by capillary bridge confinement, the morphology of the QD pixels was characterized with AFM (**Fig. R8c**), with corresponding height profiles along blue and orange solid lines shown in **Fig. R8d**. The results demonstrate that the microhole created via photolithography were successful filled with deposited QDs.

Fig. R8 a, AFM image of photoresist layer before the deposition of QDs (scale bar is 5 μm) and **b**, corresponding height profile along the blue and orange solid line. **c**, AFM

image of photoresist layer after the deposition of QDs (scale bar is 5 μm) and **d**, corresponding hight profile along the blue and orange line.

To enhance the quality of our manuscript, we have added relevant discussions in the revised version, as detailed below:

Page 7, Lines 20-23: Atomic force microscopy (AFM) images further show that the surface of the QD microstructures containing 3-F-CA is smoother (Fig. 2e and Supplementary Fig. 13), whereas the QD microstructures without 3-F-CA exhibit higher surface roughness (Supplementary Fig. 11b).

Comment 7: In the SCLC analysis, the authors appear to assume a trap-free model to extract charge carrier mobility. However, this assumption may oversimplify the actual transport behavior, as trap states are commonly present in QD-based systems. Incorporating a model that considers either a single trap level or a distribution of trap states (e.g., Gaussian disorder model) may yield a more accurate assessment of the mobility. Furthermore, the authors are encouraged to discuss whether hole injection efficiency can be estimated from the SCLC or dark-injection SCLC (DI-SCLC) measurements.

Response to comment 7: We sincerely appreciate Reviewer 2's constructive comments. Regarding the calculation of carrier mobility, the method used in our manuscript was based on established literature (*Adv. Mater.* **37**, 2413183 (2025); *Nature* **586**, 385–389 (2020); *Nature* **612**, 679–684 (2022); *Nat. Energy* **9**, 1378–1387 (2024); *Nat. Commun.* **11**, 3378 (2020)). Additionally, we recalculated it using the single-trap-level model as suggested by the reviewer. The mobility μ for each device by fitting the single trap level model (*J. Appl. Phys.* **83**, 5558-5560 (1998); *Synthetic Metals* **158**, 620-629 (2008)):

$$J = \frac{9\varepsilon\varepsilon_0\mu\theta V^2}{8L^3}$$

where J is the current density, V is the applied voltage, L is the thickness of the QD layer, ε is the relative dielectric constant, ε_0 is the vacuum permittivity and θ is the trapping factor given by

$$\theta = \left(\frac{N_v}{N_t}\right)\exp\left(\frac{-E_t}{k_B T}\right)$$

where N_v is the density of states, k_B is Boltzmann constant, E_t is the trap at energy obtained by fitting the I - V curves at different temperatures using the Arrhenius equation (**Fig. R9**). The trap density N_t is linearly proportional to trap-filled limit voltage V_{TFL} at

which a transition of I - V behavior from ohmic to TFL occurs:

$$V_{\text{TFL}} = N_t \frac{eL^2}{2\epsilon\epsilon_0}$$

The hole mobility calculated via the single-trap model is $9.35 \times 10^{-6} \text{ cm}^2 \text{ V}^{-1} \text{ s}^{-1}$ for QLED devices with long-range ordered configuration and $4.22 \times 10^{-6} \text{ cm}^2 \text{ V}^{-1} \text{ s}^{-1}$ for QLED devices with disordered configuration.

Fig. R9 a, Log-log plots of current density against voltage. **b**, Variation of $\ln(I)$ with $1/T$.

To enhance the quality of our manuscript, we have added relevant discussions in the revised version, as detailed below:

Page 7, Lines 35-36; Page 8, Lines 1-6: The electron and hole mobilities of both devices were calculated by fitting the space-charge-limited current (SCLC) region, assuming single trap level (see details of calculation in Supplementary Note 2)^{33,34}. The results show that the hole mobility ($9.35 \times 10^{-6} \text{ cm}^2 \text{ V}^{-1} \text{ s}^{-1}$, Fig. 2h) and electron mobility ($1.44 \times 10^{-4} \text{ cm}^2 \text{ V}^{-1} \text{ s}^{-1}$, Supplementary Fig. 15c) of the QLED devices with long-range ordered configuration are 2.2 times and 1.6 times higher, respectively, compared to the hole mobility ($4.22 \times 10^{-6} \text{ cm}^2 \text{ V}^{-1} \text{ s}^{-1}$, Fig. 2h) and electron mobility ($0.9 \times 10^{-4} \text{ cm}^2 \text{ V}^{-1} \text{ s}^{-1}$, Supplementary Fig. 15c) of the QLED devices with disordered configuration.

Comment 8: The manuscript reports the device operational stability in terms of T95 at 1000 nit. It would be beneficial to include additional lifetime metrics, such as T50, to provide a more comprehensive evaluation of device stability. Reporting multiple lifetime metrics not only enhances the credibility of the presented data but also facilitates more accurate comparisons with previously reported studies. In addition, given the remarkably high EQE values, independent validation through cross-laboratory measurements would further strengthen the reliability and reproducibility of the results.

Response to comment 8: We sincerely appreciate the reviewer 2's constructive suggestions. We measured the elapsed time for luminance to decay to 50% of its initial value (L_0) under different initial luminance, as illustrated in **Fig. R10a**. By fitting the equation $L_0^n T_{50} = \text{constant}$, it was inferred that the QLED with long-range ordered configuration ($n = 1.80$, **Fig. R10b**) has a T_{50} of 390 h at $1,000 \text{ cd m}^{-2}$ and the QLED with disordered configuration ($n = 1.79$, **Fig. R10c**) has a T_{50} of 163 h at $1,000 \text{ cd m}^{-2}$. Furthermore, two representative QLED devices were distributed to the following institutions for characterization: Shanghai University (device 1), and Suzhou Xingshuo Nanotech Co., Ltd (device 2). The corresponding measurement results (EQE-luminance (EQE- L) curves and current density-luminance-voltage (J - L - V) curves) have been presented in **Fig. R11**, demonstrating the excellent reproducibility of our devices.

Fig. R10 a, Luminance and time dependency characteristics curves of ordered and disordered QD-based QLEDs. Extrapolation of accelerating factor (n) for the lifetime estimation by fitting the $\text{Log}(T_{50})$ - $\text{Log}(L_0)$ data points of **b**, ordered QD-based QLED, and **c**, disordered QD-based QLED.

Fig. R11 EQE-luminance characteristics curves and current density-luminance-voltage characteristics curves of **a**, device 1 and **b**, device 2.

To enhance the quality of our manuscript, we have added relevant discussions in the revised version, as detailed below:

Page 8, Lines 34-35; Page 9, Lines 1-6: By fitting the equation $L_0^n T_{95} = \text{constant}$ ³⁶, it was inferred that the QLED with long-range ordered configuration ($n = 1.8$, Supplementary Fig. 20a) has a T_{95} of 54 h at $1,000 \text{ cd m}^{-2}$, which is 3.6 times longer than the T_{95} of 15 h for the disordered QLED ($n = 1.78$, Supplementary Fig. 20b). Additionally, T_{50} operational lifetime measurements were also conducted (Supplementary Figs. 20c and d), the T_{50} operational lifetimes were calculated to be $T_{50}@1,000 \text{ cd m}^{-2} = 390 \text{ h}$ for ordered QD-based QLED and $T_{50}@1,000 \text{ cd m}^{-2} = 163 \text{ h}$ for disordered QD-based QLED.

Response to Reviewer 3:

Comment 1: This manuscript presents a significant advancement in the field of blue quantum dot light-emitting diodes (QLEDs) by introducing an aromatic-enhanced capillary bridge confinement strategy to achieve long-range ordered blue QD arrays. This strategy was enabled by employing a ligand 3-C-FA, which introduces strong ligand-ligand interaction between small blue QDs that originally exhibit weak van der Waals interactions due to its small size/volume. By exploiting the long-range ordered QD arrays, the authors demonstrate excellent device performance with a peak external quantum efficiency (EQE) of 24.1%, a peak luminance of 101,519 cd m⁻², and an impressive T95 lifetime of 54 hours at 1,000 cd m⁻². Additionally, the authors achieve a minimum pixel size of 3 μm, which corresponds to a potential resolution exceeding 5,000 pixels per inch (PPI). The research is well-structured, and the experimental methodology is well-articulated. However, the following concerns must be adequately addressed if its publication to Nature Communications is to be considered.

Response to comment 1: We sincerely thank the Reviewer 3 for the thoughtful and encouraging assessment of our work. We are pleased that the Reviewer 3 recognizes the significance of our aromatic-enhanced capillary bridge confinement strategy for achieving long-range ordered blue QD arrays and its contribution to improving device performance. We also appreciate the reviewer's positive evaluation of the manuscript structure and methodology. In response to the specific concerns raised, we have carefully revised the manuscript and addressed each point in detail below. We are confident that these revisions have significantly improved the clarity, rigor, and overall contribution of our work, making it well-suited for publication in *Nature Communications*.

Comment 2: Major comments

Distinguishing the Role of Ligand Shortening vs. Ordering in Performance Enhancement

The key contribution of this work is that the enhanced ligand–ligand interaction improves the QD film ordering, leading to significant device performance improvements. However, an important factor to consider is that the introduction of 3-F-CA shortens the ligand length, reducing interparticle distance. It is crucial to distinguish whether the observed EQE enhancement arises predominantly from improved ordering or simply from decreased interparticle distance. A simple experiment to verify this would be to compare GISAXS measurements of randomly coated (e.g., spin-coated) films using QDs with 3-F-CA ligands (the film that yielded

EQE of 16.4% in line 151. If interparticle distance is reduced but the ordering remains disordered, and if these disordered films show a significant performance difference from structured films, it would confirm that ordering plays the primary role. Another approach would be to test a ligand with a length similar to OA but with strong π - π interactions similar to 3-F-CA.

Response to comment 2: We sincerely appreciate Reviewer 3’s constructive comments. We performed GISAXS characterization on spin-coated OA modified QD films and 3-F-CA modified QD films. As shown in **Fig. R12a** and **b**, the spin-coated films exhibited no discernible diffraction features, indicating disordered assembly. In contrast, the capillary bridge-confined assembled 3-F-CA modified QD microstructures demonstrated long-range ordered structures (**Fig. R12c**). When integrated into QLEDs, the spin-coated 3-F-CA modified QD-based QLED devices exhibited an 8.7% increase in EQE compared to OA modified QD-based QLED devices (**Figs. R12d** and **e**), suggesting that 3-F-CA modification reduces interdot spacing and significantly enhances device performance (*Nature* **586**, 385-389 (2020)). However, the capillary bridge-confined assembled 3-F-CA-modified QD-based QLED devices achieved a further 7.7% EQE improvement over their spin-coated counterparts, ultimately reaching 24.1% (**Fig. R12f**) (*Nat. Commun.* **16**, 4257 (2025)). Thus, the achievement of the highest performance primarily stems from the synergistic effect of ligand modification and long-range ordered QD assembly.

Fig. R12 a, GISAXS pattern of spin-coated OA-modified QDs. **b**, EQE- luminance curves of spin-coated OA modified QD-based QLED. **c**, GISAXS pattern of randomly coated 3-F-CA-modified QDs. **d**, EQE- luminance curves spin-coated 3-F-CA modified QD-based QLED. **e**, GISAXS pattern of long-range ordered 3-F-CA-modified QDs. **f**,

EQE- luminance curves of assembled 3-F-CA modified QD-based QLED.

Comment 3: Applicability to Green and Red QLEDs: The authors focus exclusively on blue QDs, explaining that blue QDs have inherently weaker van der Waals forces due to their smaller size. However, it is unclear whether the same approach could be extended to green and red QDs. Does the introduction of 3-F-CA have marginal effect/weak effect on green/red QDs simply because their van der Waals interactions are already strong enough to induce ordering? Could 3-F-CA still contribute by further reducing interparticle distance in green/red QDs, thereby enhancing performance? Experimental verification using green and red QDs would clarify whether this strategy is beneficial beyond blue QDs.

Response to comment 3: We sincerely appreciate Reviewer 3's insightful question. As is well established, the low efficiency and lack of effective patterning strategies for blue QLEDs remain major challenges hindering the commercialization of full-color QD displays. Therefore, our study has been primarily focused on addressing these critical bottlenecks in blue QLEDs, particularly in the context of patterned device fabrication. In contrast, both red and green QLEDs have already demonstrated performance levels that surpass commercial requirements (**Fig. R13**) (*Science Bulletin* **70**, 905-913 (2025); *Nat. Commun.* **15**, 5161 (2024)), which is why we did not carry out detailed investigations on these systems in this work. In fact, even without spatial confinement, red and green QDs can form ordered structures via spin-coating (**Fig. R14**) (*Nat. Photon.* **18**, 186-191 (2024); *Nano Energy* **140**, 110982 (2025)).

[FIGURE REDACTED]

Fig. R13 a, Current density-luminance-voltage curves and **b**, EQE-luminance of red QLED (red QLED: *Science Bulletin* **70**, 905-913 (2025)). **c**, Current density-luminance-voltage curves and **d**, EQE-current density of green QLED (green QLED: *Nat. Commun.* **15**, 5161 (2024)).

[FIGURE REDACTED]

Fig. R14 a, GIWAXS pattern of red QD film. **b**, EQE and current efficiencies versus luminance of red QLED (red QLED: *Nat. Photon.* **18**, 186-191 (2024)). **c**, GIWAXS pattern of green QD film. **d**, EQE and current efficiencies versus luminance of green QLED (green QLED: *Nano Energy* **140**, 110982 (2025)).

Comment 4: Quantifying Interaction Strength for Long-Range Ordering

The authors perform DFT calculations to estimate ligand interaction strength, but it remains unclear how much interaction strength is required to achieve long-range ordering. Could the authors provide a quantitative threshold for interaction energy required for long-range ordering? How do the interaction energies of green and red QDs compare, and do they naturally meet this threshold even without introducing 3-F-CA?

Response to comment 4: We appreciate Reviewer 3's insightful comments. In capillary bridge-confined assembly experiments, we observed a notable influence of QD size on long-range ordering: when the size of OA-capped QDs exceeds 10 nm, long-range ordered structures spontaneously form without introducing 3-F-CA ligands (**Fig. R15**). This phenomenon suggests the existence of a critical threshold for inter-QD interaction strength, which is size-dependent. Full-scale simulations of entire QD systems comprising thousands of atoms, along with comparative analyses of QDs with different

atomic compositions (e.g., blue/green/red-emitting QDs), exceed the scope of currently available computational resources. While we cannot quantify specific threshold values, the size of QDs in the visible spectrum can theoretically be estimated.

Fig. R15 SEM images of QDs with different sizes (scale bar is 20 nm).

Comment 5: QD Size Considerations in Other Systems

The manuscript states that blue QDs used here have a diameter of ~ 8 nm. However, commonly used green/red QDs (e.g., InP) have similar diameters. Would applying 3-F-CA to InP QDs (or other large QDs) also lead to enhanced performance?

Response to comment 5: We sincerely appreciate the constructive comments provided by Reviewer 3. We conducted experiments on green InP QDs (InP/ZnSe/ZnS, 529 nm) with an average diameter of 7.92 ± 0.82 nm, synthesized following established protocols (*Nature* **635**, 854–859 (2024); *Light Sci. Appl.* **11**, 162 (2022); *Adv. Sci.* **9**, 2200959 (2022)) (**Fig. R16**). After the 3-F-CA ligand exchange, we fabricated QLED devices using the spin-coating method. As shown in **Fig. R17**, the 3-F-CA modification only brought limited performance improvements (peak EQE increased from 4.3% to 6.3%, and maximum luminance rose from 2,346 cd m^{-2} to 7,613 cd m^{-2}). We speculate that the morphological differences between irregular tetrahedral InP QDs and near-spherical Cd-based QDs hinder ordered packing. Further systematic studies are underway to investigate the mechanisms regulating the self-assembly behavior of InP QDs and their impact on device performance.

Fig. R16 a, UV-Vis absorption and PL spectra of synthesized InP QDs. **b**, TEM image and size distribution histogram of synthesized InP QDs (scale bar is 50 nm).

Fig. R17 a, EQE-current efficiency-luminance curves and **b**, Current density-luminance-voltage curves of OA and 3-F-CA modified InP QD-based QLED.

Comment 6: Surface Binding Energy Differences Between OA and 3-F-CA

The authors present binding energy calculations showing a significant difference between OA and 3-F-CA, despite both ligands binding via carboxylate groups. What accounts for this large discrepancy in binding energy? Does π - π stacking contribute directly to surface binding stability, or is another mechanism at play?

Response to comment 6: We thank the Reviewer 3 for raising this critical question. We think that the observed disparity in binding energies between these two carboxylate-based ligands can be attributed to two primary factors (*Nature* **629**, 586–591 (2024)):

(1) Steric hindrance effects: The distinct size of OA and 3-F-CA result in varying steric hindrance between QDs modified by these two kinds of molecules, as evidenced by the optimized configurations shown in **Fig. R20a**.

(2) Charge distribution: OA exhibits low polarity with uniform surface charge density, whereas 3-F-CA features an extended conjugated structure and ununiform charge redistribution due to the strong electron-withdrawing nature of fluorine atoms which can be derived from the electrostatic potential (ESP) maps of OA and 3-F-CA (**Fig. R20b**).

In the contribution mechanism of π - π interactions to surface binding stability, computational software considers the electronic aggregation state of the entire configuration and optimizes atomic arrangement based on the distribution of these electrons, calculating the total energy of the system, i.e., the sum of atomic potential

and kinetic energy. Here, π - π interactions primarily manifest as electron density redistribution resulting from molecular orbital overlap. This redistribution alters the local charge environment of relevant atoms, thereby indirectly influencing their spatial positions and potential energy surface characteristics. Specifically, when π - π stacking occurs between aromatic systems, the delocalization properties of their electronic aggregation states drive the system to adopt specific conformational arrangements during energy optimization. However, the stability of such configurations inherently stems from the optimization of overall electron distribution rather than π - π stacking itself as an independent energy term (*Separation and Purification Technology* **355**, 129678 (2025)). Thus, the role of π - π stacking tends to indirectly affect binding energy by modulating charge polarization and orbital interactions between atoms, rather than directly contributing significant stabilization energy.

Fig. R20 **a**, Adsorption model of OA and 3-F-CA modified QD. **b**, ESP maps of OA and 3-F-CA.

Comment 7: Quantification of Ligand Exchange and Interaction Strength Modulation

The authors quantify OA reduction by 39% upon ligand exchange. However, there is no quantitative assessment of 3-F-CA incorporation. Can the amount of 3-F-CA on the surface be measured? Would further increasing 3-F-CA concentration enhance interparticle interactions and improve ordering?

Response to comment 7: We sincerely appreciate Reviewer 3's constructive suggestions. To determine the surface coverage of 3-F-CA on QDs, we conducted X-ray photoelectron spectroscopy (XPS) characterization (*Nature* **586**, 385 – 389 (2020); *Adv. Funct. Mater.* 2501770, (2025), <https://doi.org/10.1002/adfm.202501770>). The Zn

and F signal peaks were integrated, as shown in **Fig. R21**. The Zn 2*p* orbital signals originate from the outermost ZnS layer of the QDs (orange peak) and Zn bound to carboxyl (-COOH) groups (green peak).

Fig. R21 a, XPS spectra of Zn 2*p* orbit of the 3-F-CA modified QD film. **b**, XPS spectra of F 1*s* orbit of the 3-F-CA modified QD film.

The integrated peak areas of F and Zn were used to quantify the relative number of 3-F-CA ligands. The Zn and F peaks were integrated, and their atomic ratios were calculated by normalizing their XPS sensitivity factors. The atomic ratio was determined using the following formula:

$$R_{X:Y} = \frac{\frac{A_X}{S_X}}{\frac{A_Y}{S_Y}}$$

where A_X and A_Y are the XPS peak areas of elements X and Y obtained from spectral integration, and S_X and S_Y are the sensitivity factors for elements X and Y. Using the Zn signal associated with carboxylic acid (-COOH) groups in the QDs as a reference (**Table R2**), the normalized ratio of F was determined to be 0.84.

Table R2. Atomic ratios of surface elements in the 3-F-CA modified QD film derived from XPS results.

Element	Area	RSF	Atomic ratio
Zn (Zn-O)	46,669	28.72	1
F	6,044	4.43	0.84

Moreover, in the surface modification process of QDs, the original long-chain organic ligand OA plays a critical role in maintaining the monodispersity of QDs in solution. As a short-chain ligand, the introduction of 3-F-CA can modulate the interactions between QDs. However, when the concentration of 3-F-CA is excessively

increased, the interdot interactions are significantly enhanced, leading to irreversible aggregation. This aggregation behavior induces exciton energy transfer and an increase in surface defect states, ultimately resulting in a decline in fluorescence quantum yield and deterioration of spectral properties. Therefore, optimizing the concentration of 3-F-CA requires a balance between enhancing interdot interactions and maintaining colloidal stability, as merely increasing ligand concentration is detrimental to the formation of long-range ordered QD microstructures.

To enhance the quality of our manuscript, we have added relevant discussions in the revised version, as detailed below:

Page 5, Lines 33-36; Page 6, Lines 1: To determine the ligand density of 3-F-CA on the QDs surfaces, deconvolution analysis of the X-ray photoelectron spectroscopy (XPS) peaks was performed (Supplementary Fig. 7). Using the Zn signal associated with carboxylic acid (-COOH) groups in the QDs as a reference (Supplementary Table 1), the normalized ratio of F was determined to be 0.84.

Comment 8: Line 174, the manuscript reports differing results at low vs. high ligand concentrations. A more detailed explanation of why these variations occur would help improve clarity.

Response to comment 8: We appreciate Reviewer 3 for their meticulous comments. We have revised the relevant content of the manuscript as follows:

Page 6, Lines 24-35; Page 7, Lines 1-2: Subsequently, QD arrays were prepared using QD solutions with different concentrations (2, 5, 10, and 15 mg mL⁻¹). The fluorescence micrographs of the obtained QD arrays are shown in Fig. 2b, with schematic illustrations of the assembled QD microstructures displayed in the upper-left corner of each subfigure. As the solvent continued to evaporate and the TPCL shrank, QDs preferentially reached saturation and deposited in the TPCL around the micropores. At a low QD concentration (2 mg mL⁻¹), the QDs in the solution were insufficient, resulting in incomplete microstructure formation, manifested as insufficient QD deposition in the central region. As the QD concentration increased, the unfilled central area gradually decreased (5 mg mL⁻¹) until an appropriate concentration achieved complete filling (10 mg mL⁻¹). Conversely, at an excessively high concentration (15 mg mL⁻¹), more QDs remained in the central region, producing a brighter area in the center. Based on the above investigations, we successfully fabricated large-area blue QD arrays using 3-F-CA modified QD solution with a concentration of 10 mg mL⁻¹.

Comment 9: The manuscript reports a surprising 7°C temperature difference based on QD ordering, which is surprisingly high (although I do not have much experience on such measurement). If ordering significantly alters thermal conductivity to that extent, can you measure the thermal conductivity of these films and directly confirm if QD films simply changing the particle ordering can yield such a dramatic change in thermal conductivity? having different ordering?

Response to comment 9: We sincerely appreciate the constructive comments from Reviewer 3. The significant temperature difference observed in this study primarily stems from altered energy conversion pathways induced by QD ordering, rather than direct changes in thermal conductivity. In disordered QD systems, charge leakage occurs and accumulates at interfaces, enhancing non-radiative recombination, that converts electrical energy into heat. In contrast, the formation of ordered structures effectively suppresses charge leakage, promoting energy release via radiative recombination in the form of photons. This transition from non-radiative to radiative recombination fundamentally reduces heat generation within the system, thereby producing the pronounced temperature difference (*Nat. Nanotechnol.* **18**, 1168–1174 (2023); *Nano Lett.* **23**, 6689-6697 (2023)).

Comment 10: Verification of PPI Calculation for 3 µm Patterns

The manuscript claims a resolution exceeding 5,000 PPI for 3 µm pixel sizes. However, assuming an RGB subpixel structure, a full pixel would typically be larger than 3 µm (likely ~10 µm per pixel – 3 µm blue + 3 µm green + 3 µm red). How was the 5,000 PPI figure calculated? I understand that this calculation depends on the design of the subpixels as well. It would be nicer to explain what assumptions were made regarding pixel layout.

Response to comment 10: We sincerely appreciate the reviewer's thoughtful comments. The resolution for full-color RGB pixel arrays is influenced by the size and spacing of individual RGB subpixels. However, our study focuses on high-resolution blue-patterned QLEDs. The resolution calculation we used follows standard practices from recent literature on patterned monochromatic QLEDs (e.g., *Nat. Photon.* **16**, 297–303 (2022); *Nat. Photon.* **18**, 1105–1112 (2024); *Adv. Funct. Mater.* **35**, 2413811 (2025); *Adv. Mater.* **35**, 2303329 (2023)).

In our revised manuscript, we have clarified the resolution calculation formula and the corresponding parameters. Using a pixel size of 3 µm and pixel spacing of 2 µm, the calculated resolution is 5,080 PPI, which aligns with our claim of "exceeding 5,000 PPI". This update ensures greater clarity and consistency with existing studies in the

field.

Page 6, Lines 3-7 in the Supporting Information: Resolution is conventionally quantified in pixels per inch (PPI), representing the number of individual pixels contained within one inch:

$$\text{PPI} = \frac{\sqrt{X^2 + Y^2}}{Z}$$

where X is the number of horizontal pixels, Y is the number of vertical pixels, and Z is the diagonal size of the screen (inch).

Comment 11: Definition of Active Area in Device Performance Measurements

The manuscript suggests that photoresist was used as a pixel defining layer. How was the active area defined for current density and luminance calculations?

Response to comment 11: We sincerely thank Reviewer 3 for the valuable question. The active area used for patterned QLED performance test (involving parameters such as current density, luminance, EQE, and current efficiency) has been described in Supplementary Note 7 of the previous version of Supporting Information, including the calculation formula, corresponding parameters, and their physical meanings. We have refined this section to provide clearer description, with the following modifications made:

Page 5, Lines 3-10 in the Supporting Information: In patterned QLED devices, the regions covered by insulating photoresist layers are electrically non-conductive, necessitating the exclusion of these areas during device performance characterization. Consequently, the active area (S_l) is calculated using the following formula:

$$S_l = S \times \frac{S_a}{S_b}$$

where S denotes the total device area (i.e., the overlapping region between cathode and anode), which was fixed at 3.0 mm^2 in our experimental configuration; S_a represents the area of individual pixel; S_b denotes the area of individual repeating unit.